# STLDM: Spatio-Temporal Latent Diffusion Model for Precipitation Nowcasting

**Shi Quan Foo**                                                                                  *sqfoo@connect.ust.hk*
*The Hong Kong University of Science and Technology*

**Chi-Ho Wong**                                                                              *chwongcc@connect.ust.hk*
*The Hong Kong University of Science and Technology*

**Zhihan Gao**                                                                           *zhihan.gao@connect.ust.hk*
*The Hong Kong University of Science and Technology*

**Dit-Yan Yeung**                                                                                 *dyyeung@cse.ust.hk*
*The Hong Kong University of Science and Technology*

**Ka-Hing Wong**                                                                                 *khwong@hko.gov.hk*
*Hong Kong Observatory*

**Wai-Kin Wong**                                                                                 *wkwong@hko.gov.hk*
*Hong Kong Observatory*

**Reviewed on OpenReview:** *https://openreview.net/forum?id=f4oJwXn3qg*

## Abstract

Precipitation nowcasting is a critical spatio-temporal prediction task for society to prevent severe damage owing to extreme weather events. Despite the advances in this field, the complex and stochastic nature of this task still poses challenges to existing approaches. Specifically, deterministic models tend to produce blurry predictions while generative models often struggle with poor accuracy. In this paper, we present a simple yet effective model architecture termed **STLDM**, a diffusion-based model that learns the latent representation from end to end alongside both the Variational Autoencoder and the conditioning network. STLDM decomposes this task into two stages: a deterministic *forecasting* stage handled by the conditioning network, and an *enhancement* stage performed by the latent diffusion model. Experimental results on multiple radar datasets demonstrate that STLDM achieves superior performance compared to the state of the art, while also improving inference efficiency. The code is available in `https://github.com/sqfoo/stldm_official`.

## 1 Introduction

Precipitation nowcasting is a short-term prediction task for precipitation events over a specific region, based on weather data such as radar and satellite observations. Accurate and timely nowcasting is crucial to society, as it enables us to take preventive actions for mitigating potential economic loss and other adverse impacts due to extreme weather. Traditionally, meteorologists utilized algorithmic methods such as optical-flow methods (Pulkkinen et al., 2019) and the guidance from numerical weather prediction (NWP) models on this nowcasting task.

With the emergence of deep learning, data-driven models have been extensively explored for modeling the spatio-temporal patterns of precipitation events. Despite their lack of interpretability, these deep learning approaches often outperform traditional methods in terms of accuracy and efficiency. Broadly, these approaches can be categorized into two main research categories: video prediction (Shi et al., 2015; 2017; Gao

et al., 2022b), which models the 4D spatio-temporal trend with a ground truth observation for performance evaluation; and video generation (Zhang et al., 2023; Leinonen et al., 2023; Gao et al., 2023), which adopts generative models to synthesize the target data distributions with less consideration to the alignment to the ground truth and more emphasis on the visual fidelity.

Recent works highlight the challenges posed by the stochastic nature of precipitation nowcasting due to the inherent unpredictability of open systems. Deterministic models, in particular, tend to capture the global motion trend well with the missing of details at the micro-level, resulting in blurry predictions over longer lead times. This leads to the difficulty in practical forecasting operations (Ravuri et al., 2021). On the other hand, generative models are capable of modeling micro-level weather phenomena through simulating the data distribution, which tolerates the nature of stochasticity, thereby producing realistic and sharp forecasts (Zhang et al., 2023; Leinonen et al., 2023; Gao et al., 2023). However, they often suffer from low accuracy in predicting large-scale weather events. In summary, both these approaches could only achieve either high accuracy (deterministic models) or high visual quality (generative models) as shown in Figure 1.

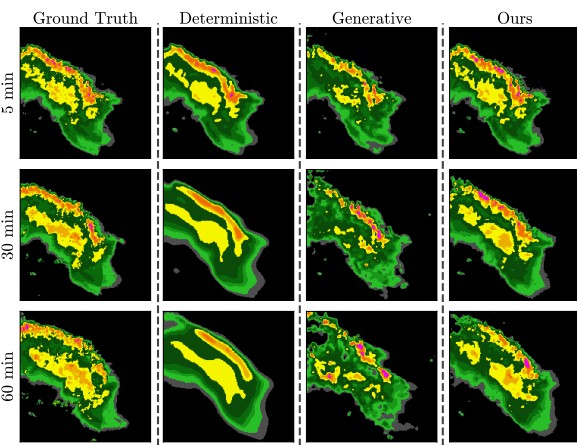

Figure 1: This demonstrates that deterministic models result in blurry predictions while generative models suffer from the issue of inaccurate predictions. Our proposed STLDM is capable of forecasting accurate predictions while maintaining a nice appearance.

In this paper, we first re-formulate the precipitation nowcasting task into two sequential subtasks: **Forecasting** and **Enhancement** based on the observations above. First, a Translator with the same architecture as deterministic models is implemented to perform the forecasting task, i.e., roughly forecast the upcoming precipitation events, $\overline{Y}$. The objective here is to capture the approximate global motion trend of the future. Then, a diffusion model is used to refine the first estimation by the Translator, $\overline{Y}$, by introducing it as the conditional variable such that the global motion trend of generated samples is constrained well. To accelerate the inference process, we employ this sampling process in the latent space.

Therefore, we propose and present a novel and simple Spatio-Temporal Latent Diffusion Model – **STLDM** for these re-formulated objectives, effectively handling the stochasticity in precipitation nowcasting. STLDM consists of three modules: a Variational AutoEncoder, a Translator (aka Conditioning Network), and a Latent Denoising Network. Furthermore, we train STLDM in a manner from end to end to encourage cooperation among all modules. The experimental results show that STLDM is capable of achieving the state-of-the-art performance on both pixel accuracy and visual fidelity, outperforming most diffusion-based models. The contributions of this work are summarized as follows:

- We re-formulate the precipitation nowcasting task into: Forecasting and Enhancement in sequence. Based on this, we propose STLDM, a simple yet efficient model utilizing the Latent Diffusion Model.
- To the best of our knowledge, this is the first work that trains a Latent Denoising Network alongside a Variational AutoEncoder and a Conditioning Network for the precipitation nowcasting task.
- Our STLDM achieves state-of-the-art performance on multiple real-life radar echo datasets across most evaluation metrics while offering faster sampling speeds compared to other diffusion-based models.

## 2 Related Works

### 2.1 Precipitation Nowcasting as a Spatio-Temporal Task

Precipitation nowcasting is commonly interpreted as a spatio-temporal predictive task to predict the next $N$ output frames, $\hat{Y}_{1:N}$, given $M$ input frames, $X_{1:M}$. It is formulated as:

$$\underset{\hat{Y}_1,...,\hat{Y}_N}{\arg\max}\, p(\hat{Y}_1, ..., \hat{Y}_N \mid X_1, ..., X_M) \tag{1}$$

Based on this formulation, various deep learning models that leverage spatio-temporal features have been proposed. ConvLSTM (Shi et al., 2015), which integrates convolution layers into LSTM cells in an encoder-forecaster architecture, is the first method proposed for this task. Later, PredRNN (Wang et al., 2017) introduced a novel structure, the ST-LSTM unit, to extract spatio-temporal features and model future frames in a zigzag memory flow. Building on this foundation, several modifications have been proposed, including Memory In Memory (MIM) (Wang et al., 2019), the gradient highway (Wang et al., 2018), and reversed scheduled sampling (Wang et al., 2023).

Following the success of Transformers and their attention mechanisms in natural language processing (NLP) tasks (Vaswani et al., 2017) and vision processing tasks (Dosovitskiy et al., 2021), several Transformer-based models have been deployed to capture the long-term spatio-temporal features of this nowcasting task. For instance, Cuboid attention in Earthformer (Gao et al., 2022b) and Feature Extraction Balance Module in Rainformer (Bai et al., 2022) are proposed to extract and model both the global and local rainfall features.

In parallel, CNN-based models have also been explored for spatio-temporal modeling, like SimVP (Gao et al., 2022a) and TAU (Tan et al., 2023a). Both works adopt a U-Net-like structure, i.e., Encoder-Translator-Decoder with Spatial Encoder and Decoder. SimVP and TAU introduced Inception modules and Temporal Attention Unit as translators, respectively, to learn the temporal evolution. These works are primarily composed of convolution operations, achieving efficiency and remarkable performance.

However, neither of these works can produce sharp predictions for long lead times due to the stochastic nature of this task. To address this issue, several loss functions were proposed as alternatives to the conventional L2 loss, such as SSL (Chen et al., 2020) and FACL (Yan et al., 2024). Meanwhile, probabilistic models have been employed to capture the spatio-temporal features by estimating the conditional distribution of the future frames, thereby enabling ensemble predictions with high visual fidelity, such as GANs (Ravuri et al., 2021; Chang et al., 2022; Zhang et al., 2023) and variational autoencoders (VAEs) (Denton & Fergus, 2018; Franceschi et al., 2020). However, these models are often criticized for their training instability and potential for mode collapse.

### 2.2 Diffusion-based Models

Diffusion models (DMs) (Ho et al., 2020) have achieved high visual fidelity in image generation (Saharia et al., 2022; Ramesh et al., 2022) and video generation (Ho et al., 2022; Yang et al., 2023; Voleti et al., 2022), and are increasingly being applied in atmospheric forecasting(Price et al., 2023) and downscaling(Mardani et al., 2023; Wan et al., 2024). Unlike GANs, DMs, which are optimized with a likelihood objective (Kingma & Gao, 2023), do not suffer from mode collapse. However, they are computationally expensive and have longer inference times. Several techniques have been proposed to accelerate the sampling process, such as DDIM (Song et al., 2021) and progressive distillation (Salimans & Ho, 2022).

Applying the denoising process in the latent space, Latent Diffusion Model (LDM), also reduces the heavy demand of computational resources. LSGM (Vahdat et al., 2021), the first LDM with an end-to-end training scheme for both VAE and LDM, achieved state-of-the-art performance in image generation tasks and provided a solid theoretical analysis. Later, follow-up works (Rombach et al., 2022) decomposed this framework into a two-stage process: pre-training a VAE followed by the training of LDM, showcasing its effectiveness in conditional generation tasks with lower computational demands.

LDCast (Leinonen et al., 2023) employs Adaptive Fourier Neural Operators (AFNOs) in a conditional LDM to predict the evolution of radar echo via the denoising process. Similarly, PreDiff (Gao et al., 2023)

incorporates prior knowledge through a knowledge-control network to align the output with prior knowledge, while replacing the U-Net-style architecture with Earthformer to capture complex spatio-temporal features. These works adopt the two-stage training framework described above.

Although these diffusion-based models could produce predictions with high visual quality, they often suffer from the issue of low accuracy. To address this, recent works have attempted to achieve both high visual quality and high accuracy via the cooperation between deterministic models and DMs. DiffCast (Yu et al., 2024) treats this nowcasting task as a combination of trend prediction (handled by deterministic models) and local stochastic variation (handled by DMs); while CasCast (Gong et al., 2024) introduces a cascaded modeling approach that combines deterministic models with Casformer.

In summary, DiffCast runs the denoising process in the pixel space, which leads to a longer inference time. Conversely, LDCast, PreDiff, and CasCast deploy DMs in the latent space with a pre-trained VAE. This approach is proven to be computationally efficient, but the independence to train every module limits the model's generative capability, as every module is trained with the corresponding objectives. End-to-end training of all components may further improve performance.

## 3 Methodology

In this section, we first provide the background of diffusion models. Then, we revisit and reformulate the objective of the precipitation nowcasting task. Lastly, we propose and present a Spatio-Temporal Latent Diffusion Model – STLDM in detail, including its training loss function and each component in STLDM.

### 3.1 Preliminary

#### 3.1.1 Diffusion Models

Diffusion Models (DMs) (Ho et al., 2020) learn the data distribution, $p(x)$, by modeling the reverse diffusion process from Gaussian noise, $x^T$, using corrupted samples, $x^t$, with their corresponding diffusion step, $t$. The forward diffusion process, which gradually adds noise from $t = 1$ to $t = T$, is defined as:

$$q(x^{t+1}|x^t) = \mathcal{N}(x^t; \sqrt{1 - \beta_t}x^t, \beta_t I), \tag{2}$$

where $\beta_t \in (0, 1)$ increases monotonically over $t$.

The reverse diffusion process iteratively removes noise, starting from pure Gaussian noise, $x^T$, and is formulated as:

$$p_\theta(x^{t-1}|x^t) = \mathcal{N}(x^{t-1}; \mu_\theta(x^t, t), \sigma_t^2 I), \tag{3}$$

with the parameterized posterior mean function, $\mu_\theta$ defined as:

$$\mu_\theta(x^t, t) = \frac{1}{\sqrt{\alpha_t}}(x^t - \frac{\beta_t}{\sqrt{1 - \bar{\alpha}_t}}\epsilon_\theta(x^t, t)), \tag{4}$$

where $\alpha_t = 1 - \beta_t$, $\bar{\alpha}_t = \prod_{s=1}^{t} \alpha_s$ and $\epsilon_\theta(x^t, t)$ is a trainable denoising function that estimates the noise at the diffusion step, $t$. The network parameters, $\theta$, are optimized by minimizing the diffusion loss, $\mathcal{L}_{\text{diffusion}}$, as formulated as:

$$\mathcal{L}_{\text{diffusion}} = \mathbb{E}_{x^0 \sim q(x^0), \epsilon \sim \mathcal{N}(0,\mathbf{I}), t \sim \mathcal{U}(1,T)}\left[\gamma(t)||\epsilon - \epsilon_\theta(\sqrt{\bar{\alpha}_t}x^0 + \sqrt{1 - \bar{\alpha}_t}\epsilon, t)||^2\right], \tag{5}$$

where $\gamma(t)$ is the weighting function.

#### 3.1.2 Classifier-Free Guidance

To balance the trade-off between sample quality and diversity during generation, guidance is introduced during sampling to ensure that the sample generated by the diffusion model is constrained with the given

conditional variables, $c$, such as class labels and text prompts. Inspired by GANs, Classifier Guidance (Dhariwal & Nichol, 2021) is initially implemented by incorporating the gradient of a trained classifier, $\phi$ into the diffusion score during sampling as follows:

$$\tilde{\epsilon}_{\theta,\phi}(x_t, c) = \epsilon_\theta(x_t, c) + w\sqrt{1 - \bar{\alpha}_t}\nabla_{x_t}p_\phi(c|x_t), \tag{6}$$

which modifies the sampling distribution to: $\tilde{p}(x_t|c) \propto p(x_t,|c)p(c|x_t)^w$ with the guidance strength, $w$.

According to the Bayes' Theorem, we could rewrite $p(c|x_t) \propto p(x_t|c)/p(x_t)$, thereby the sampling distribution could be further modified as: $\tilde{p}(x_t|c) \propto p(x_t,|c)\big[p(x_t)/p(x_t|c)\big]^{-w}$. With this intuition, Classifier-Free Guidance (CFG) (Ho & Salimans, 2022) was proposed as an alternative without an external classifier. Instead, an "implicit classifier" is achieved by jointly training a conditional diffusion model, $p(x_t|c)$, and unconditional diffusion model, $p(x_t)$. CFG is achieved by modifying the diffusion score, $\tilde{\epsilon}_\theta(x^t, c)$, for every sampling timestep, $t$, as:

$$\tilde{\epsilon}_\theta(x^t, c) = \epsilon_\theta(x^t, c) - w(\epsilon_\theta(x^t, \phi) - \epsilon_\theta(x^t, c)), \tag{7}$$

where the null sign $\phi$ indicates the unconditional case.

## 3.2 Proposed Approach and Details

In this part, we first reformulate the Precipitation Nowcasting task defined in Equation 1 and derive the corresponding loss function for the proposed STLDM. Later, we describe each component of the proposed STLDM in detail.

### 3.2.1 Task Reformulation

With introducing the intermediate variables, $\overline{Y}_{1:N}$, the conditional probability, $p(\hat{Y}_{1:N}|X_{1:M})$, in Equation 1 could be rewritten as:

$$p(\hat{Y}_{1:N}|X_{1:M}) = \int p(\overline{Y}_{1:N}|X_{1:M})p(\hat{Y}_{1:N}|\overline{Y}_{1:N}, X_{1:M})d\overline{Y}_{1:N}, \tag{8}$$

where $\hat{Y}_{1:N}$ and $\overline{Y}_{1:N}$ represent the decoded radar frames predicted by the Latent Denoising Network and Translator, respectively, with the given input radar frames, $X_{1:M}$. The details can be found in Appendix B.

The first term in Equation 8 corresponds to the Forecasting task of the first estimation, $\overline{Y}_{1:N}$, with the given input radar frames, $X_{1:M}$; while the second term represents the Visual Enhancement task conditioned on both the first estimation, $\overline{Y}_{1:N}$, and input radar frames, $X_{1:M}$. This motivates us to reformulate the precipitation nowcasting task into two sequential sub-tasks: **Forecasting** and **Enhancement**.

### 3.2.2 Derivation of Loss Function

Here, we derive the corresponding loss function of the proposed Spatio-Temporal Latent Diffusion Model (STLDM) as illustrated in Figure 2. Briefly speaking, the input radar frames, $X_{1:M}$, is first encoded by Spatial Encoder, $\mathcal{E}$, then Translator, $\Psi_\theta$, does the first estimation, $\overline{Y}_{1:N}$, and the Latent Denoising Network, $D_\theta$, further enhance its visual quality and obtain the final prediction, $\hat{Y}_{1:N}$.

We start with the conditional hierarchical VAE loss function over $L$ variational layers (Vahdat & Kautz, 2020):

$$\mathcal{L}_{\text{ELBO}} = \mathbb{E}_{q(z|x)}[-\log p(x|z)] + D_{\text{KL}}(q(z_1|x)||p(z_1)) + \sum_{l=2}^{L} D_{\text{KL}}(q(z_l|x, z_{<l})||p(z_l|z_{<l})), \tag{9}$$

We interpret STLDM as a 3-layer VAE and derive its corresponding loss function, $\mathcal{L}_{\text{ELBO}}$, by substituting: $z_1 \leftarrow z_x$, $z_2 \leftarrow \bar{z}_{1:N}$, $z_3 \leftarrow z_{1:N}^T$ and $z_4 \leftarrow z_{1:N}$, where $z_x$ is the latent representation of the radar frames and $z_{1:N}^T$ is the pure Gaussian Noise with mean of 0 and standard deviation of 1.

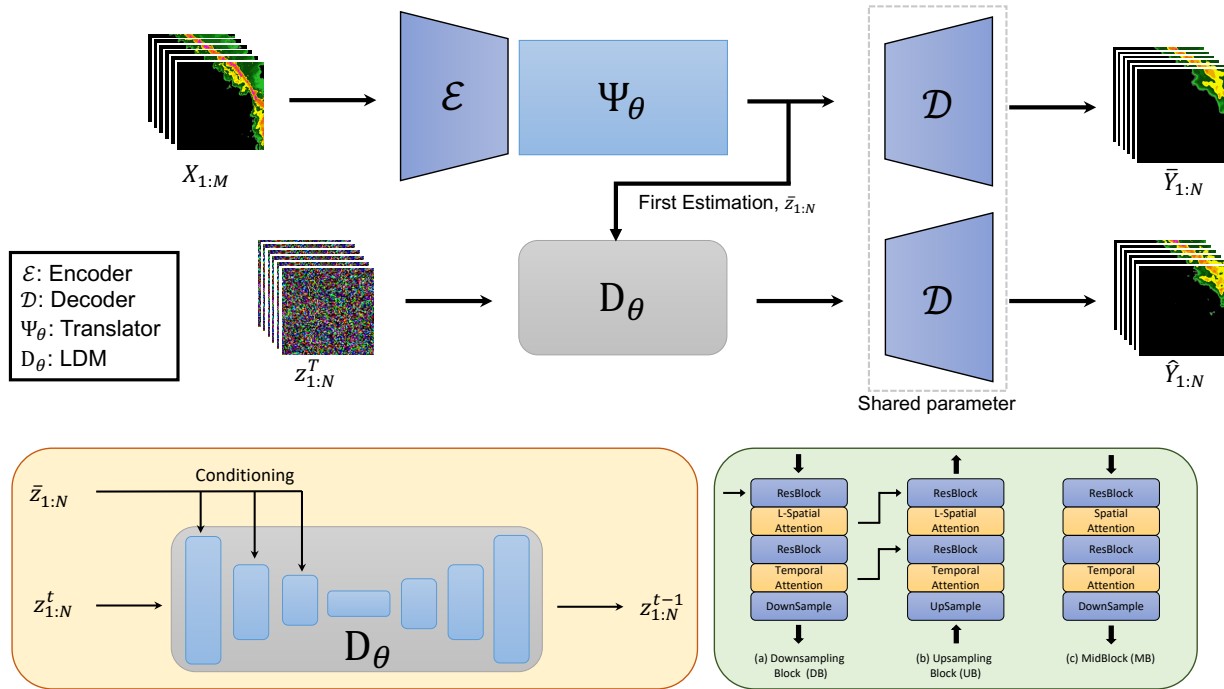

Figure 2: **Top:** Model architecture of the proposed STLDM, consisting of a Variational Autoencoder, ($\{\mathcal{E}, \mathcal{D}\}$), a Conditioning Network (aka Translator), $\Psi_\theta$, and a Spatio-Temporal Latent Denoising Network, $D_\theta$. Input radar frames are denoted as $X_{1:M}$; while the decoded outputs of both the final prediction after denoising from pure Gaussian Noise, $z_{1:N}^T$, and the first estimation, $\bar{z}_{1:N}$, are denoted as $\hat{Y}_{1:N}$ and $\overline{Y}_{1:N}$ respectively. **Bottom:** Overall architecture of $D_\theta$ (Yellow Box) and the details of its sub-modules (Green Box). "L-Spatial Attention" stands for Linearized Spatial Attention.

$$\mathcal{L}_{\text{ELBO}} = \mathbb{E}_{q(z_x|x)}[-\log p(x|z_x)] \tag{A}$$
$$+ D_{\text{KL}}(q(z_x|x)||p(z_x)) \tag{B}$$
$$+ D_{\text{KL}}(q(\bar{z}_{1:N}|x,z_x)||p(\bar{z}_{1:N}|z_x)) \tag{C}$$
$$+ D_{\text{KL}}(q(z_{1:N}^T|x,z_x,\bar{z}_{1:N})||p(z_{1:N}^T|z_x,\bar{z}_{1:N})) \tag{D}$$
$$+ D_{\text{KL}}(q(z_{1:N}|x,z_x,\bar{z}_{1:N},z_{1:N}^T)||p(z_{1:N}|z_x,\bar{z}_{1:N},z_{1:N}^T)) \tag{E}$$

Term A corresponds to a typical reconstruction loss term, $\mathcal{L}_{\text{MSE}}$, to minimize the difference between the reconstructed radar frames and the ground truth. As mentioned above, the global motion of the prediction from the deterministic model has a good alignment with the ground truth. To ensure that the final prediction, $\hat{Y}_{1:N}$, has the same global motion alignment, we propose a constraint loss term, $\mathcal{L}_C$, for regulating the first estimation, $\bar{z}_{1:N}$, which also acts as the conditioning vector on the denoising network, $D_\theta$. Hence, this loss term consists of two terms: a typical Mean Squared Error loss for VAE and a constraint loss term on Translator (aka Conditioning Network), $\Psi_\theta$.

$$\mathbb{E}_{q(z_x|x)}[-\log p(x|z_x)] = \underbrace{||X_{1:M+N} - \hat{X}_{1:M+N}||^2}_{\mathcal{L}_{\text{MSE}}} + \underbrace{||Y_{1:N} - \overline{Y}_{1:N}||^2}_{\mathcal{L}_C}, \tag{10}$$

where $X_{1:M+N}$ is the concatenation of both the input radar frames and the ground truth, and $Y_{1:N}$ is the ground truth itself, while any notations with ˆ mean the prediction from STLDM.

Term B represents a regular KL-divergence loss term for constraining the distribution of the encoded latent representation by the encoder, $\mathcal{E}$, $\mathcal{N}(\mu_\theta, \sigma_\theta)$, towards a Standard Gaussian distribution, $\mathcal{N}(0,1)$:

$$D_{\mathrm{KL}}(q(z_x|x)||p(z_x)) = D_{\mathrm{KL}}(\mathcal{N}(\mu_\theta, \sigma_\theta)||\mathcal{N}(0,1)) = \frac{1}{2}[\sigma_\theta^2 + \mu_\theta^2 - 1 - \log\sigma_\theta], \tag{11}$$

while Term C is also a KL-divergence loss term for regularizing the distribution of the first estimated latent representation, $\mathcal{N}(\bar{z}_{1:N}, \bar{\sigma}_{1:N})$, has a similar pattern as a Standard Gaussian distribution, $\mathcal{N}(0,1)$:

$$D_{\mathrm{KL}}(q(\bar{z}_{1:N}|x, z_x)||p(\bar{z}_{1:N}|z_x)) = D_{\mathrm{KL}}(\mathcal{N}(\bar{z}_{1:N}, \bar{\sigma}_{1:N})||\mathcal{N}(0,1)) = \frac{1}{2}[\bar{\sigma}_{1:N}^2 + \bar{z}_{1:N}^2 - 1 - \log\bar{\sigma}_{1:N}], \tag{12}$$

The loss term presented in Term D is represented as a Prior Loss that ensures the disrupted latent representation, $z_{1:N}^T$, follows Equation 2. In both the formulation of DDPM and LDM with pre-trained VAE, this loss term would be dropped as it is irrelevant to the denoising network itself. However, this matters for our proposed STLDM with the end-to-end tuning framework, and it is defined as:

$$D_{\mathrm{KL}}(q(z_{1:N}^T|x, z_x, \bar{z}_{1:N})||p(z_{1:N}^T|z_x, \bar{z}_{1:N})) = D_{KL}(\mathcal{N}(\sqrt{\bar{\alpha}_T}z_{1:N}, (1-\bar{\alpha}_T))||\mathcal{N}(0,1)), \tag{13}$$

where $z_{1:N}$ is the encoded latent representation of the ground truth radar frames, $Y_{1:N}$.

Following the previous work (Kingma & Gao, 2023), the last loss term stated in Equation 5 could be further defined and expressed as a general diffusion loss function for the latent denoising network:

$$D_{\mathrm{KL}}(q(z_{1:N}|x, z_x, \bar{z}_{1:N}, z_{1:N}^T)||p(z_{1:N}|z_x, \bar{z}_{1:N}, z_{1:N}^T)) = \gamma(t)||\epsilon - \epsilon_\theta(\sqrt{\bar{\alpha}_t}z_{1:N} + \sqrt{1-\bar{\alpha}_t}\epsilon, t)||^2 \tag{14}$$

In summary, the derived loss function for our proposed STLDM consists of VAE reconstruction loss, KL-divergence regularization loss, Conditioning regularization loss, Prior loss, and a diffusion loss.

### 3.2.3 Components of STLDM

Spatio-Temporal Latent Diffusion Model (STLDM) consists of three main components: a Variational AutoEncoder, $\{\mathcal{E}, \mathcal{D}\}$, a Conditioning Network (aka Translator), $\Psi_\theta$, and a Latent Denoising Network, $D_\theta$. Its details are illustrated in Figure 2. Model performance is further improved with the technique of CFG. Since CFG involves both the conditional case ($c = \bar{z}_{1:N}$) and the unconditional case ($c = \phi$) during the inference, we jointly train the latent denoising network, $D_\theta$, on both cases with the provided algorithm in (Ho & Salimans, 2022).

**Variational AutoEncoder, $\{\mathcal{E}, \mathcal{D}\}$,** learns the spatial mappings between the radar images, $X_t$, and their corresponding latent variables, $z_t$. Specifically, the encoder, $\mathcal{E}$, transforms the input radar image, $X_t$, from pixel space to the latent space; while the decoder, $\mathcal{D}$, reconstructs the predicted latent variables back from the encoded latent space to the pixel space. We denote the encoded radar image in the latent space as $z_t \sim \mathcal{E}(X_t)$ and its reconstructed radar image be $\hat{X}_t \sim \mathcal{D}(z_t)$.

**Conditioning Network/Translator, $\Psi_\theta$,** learns the temporal evolution from the input encoded input frames, $X_{1:M}$, to the target output frames, $Y_{1:N}$, in the latent space. Following the success of SimVP-V2, we employ its Gated Spatio-Temporal Attention (gSTA) module (Tan et al., 2023b), which relies solely on convolution operations as $\Psi_\theta$ to model the underlying relation between $X_{1:M}$ and $Y_{1:N}$ in the latent space. We denote this prediction as a first estimation, $\bar{z}_{1:M}$, and its decoded output as the first estimation, $\overline{Y}_{1:N}$.

**Latent Denoising Network, $D_\theta$,** is a conditional latent diffusion model that generates probabilistic predictions with conditioning on the latent prediction by $\Psi_\theta$. To effectively capture the spatio-temporal features, we decouple the spatio-temporal attention mechanism in $D_\theta$ into spatial attention and temporal attention modules. To optimize the computation, a linear variant of spatial attention (Katharopoulos et al., 2020) which reduces the complexity from $\mathcal{O}(N^2 d)$ to $\mathcal{O}(N d^2)$ is implemented for every Downsampling and Upsampling blocks, where $N$ and $d$ are the number of sequences and their projected dimension respectively.

Briefly speaking, the difference between this linearized variant and a standard attention is the operation order of the query, $Q$, key, $K$, and value, $V$, as shown below:

$$A_{\text{Standard}} = \Big(\phi(Q)\phi(K)\Big)\phi(V); A_{\text{Linear}} = \phi(Q)\Big(\phi(K)\phi(V)\Big), \tag{15}$$

STLDM is trained from end to end with the objective defined in Section 3.2.2.

## 4 Experiments

### 4.1 Experimental Settings

We evaluate the performance and effectiveness of our proposed STLDM with several deterministic models serving as baselines, together with various diffusion-based models designed for precipitation nowcasting: LD-Cast (Leinonen et al., 2023), PreDiff (Gao et al., 2023), and DiffCast (Yu et al., 2024), on three real-life radar datasets: SEVIR (Veillette et al., 2020), HKO-7 (Shi et al., 2015), and MeteoNet (Larvor et al., 2020). To alleviate the computation cost, we downscale the spatial resolution to $128 \times 128$ while preserving their temporal dimension.

#### 4.1.1 Dataset

**SEVIR**    (Veillette et al., 2020) is a curated and spatial-temporally aligned dataset that captures the weather events consisting of five different modalities in the US from 2017 to 2019. Each event consists of an image sequence spanning four hours with a time interval of 5 minutes, covering the region with the geographical size of $384\text{km} \times 384\text{km}$ in the US. The data range of the frames is set to $[0 - 255]$. Following previous work (Gao et al., 2022b), we specifically select the Vertically Integrated Liquid (VIL) channel and formulate the task as predicting the next 12 frames (60 minutes) given 13 input frames (65 minutes). We span the data collected from June to December 2019 as the test set, while the remaining is the training set.

**HKO-7**    (Shi et al., 2015) is a meteorological dataset containing sequences of observed Constant Altitude Plan Position Indicator (CAPPI) radar reflectivity at an altitude of 2km, covering the region with a radius of 256km centered at Hong Kong. The data are collected from 2009 to 2015 with a time interval of 6 minutes. The data range of the frames is set to $[0 - 255]$. Following previous work (Yan et al., 2024), we formulate this task as predicting the future radar echoes up to 2 hours (20 frames) based on the past 30 minutes (5 frames). We sample the collected data from 2009 to 2014 as the training set, while the rest is allocated to the test set.

**MeteoNet**    (Larvor et al., 2020) is an open-source meteorological dataset that consists of both satellite and radar observations with a 5-minute interval in France. The data covers geographical areas: the Northwestern and Southeast quarters of France, with the observation size of $550\text{km} \times 550\text{km}$ from 2016 to 2018. The data range of the frames is set to $[0 - 70]$. Like the HKO-7 dataset, we formulate this task to forecast the next 20 frames (100 minutes) of radar echoes based on the provided 5 frames (25 minutes) of radar echoes. Following previous work (Yu et al., 2024), we select the data specifically from Northwestern France and filter out the noisy precipitation events. The data collected from June to December 2018 serves as the test set, while the rest are used as the training set.

#### 4.1.2 Evaluation

Following previous works (Gao et al., 2023; Yu et al., 2024; Yan et al., 2024), various commonly used forecasting skill scores, such as the Critical Success Score (CSI) and the Heidke Skill Score (HSS), are reported to evaluate the performance of the models. CSI, also known as Intersection-over-Union (IoU), measures how accurate the model prediction is after binarizing pixels of both prediction and observation with a specific threshold. The reported CSI is averaged over multiple selected thresholds, i.e., $\{16, 74, 133, 160, 181, 219\}$ for SEVIR, $\{84, 117, 140, 158, 185\}$ for HKO-7, and $\{12, 18, 24, 32\}$ for MeteoNet. With the toleration of spatial deviations, averaged CSIs with pooling sizes of 4 and 16, which correspond to medium and large spatial

Table 1: Performance comparison on multiple precipitation nowcasting benchmarks: SEVIR, HKO-7, and MeteoNet. The best score among all models is highlighted in **bold**, while the best score among the probabilistic models, including ours, is underlined. The inference time, $T_{\text{sample}}$ (in seconds), on a single RTX3090 GPU and the sampling steps, $N$, of those diffusion models are reported as well.

| Dataset | Model | Metrics | | | | | | $T_{\text{sample}}$ | $N$ |
|---|---|---|---|---|---|---|---|---|---|
| | | SSIM↑ | LPIPS↓ | CSI-m↑ | $CSI_4$-m↑ | $CSI_{16}$-m↑ | HSS↑ | | |
| SEVIR | ConvLSTM | 0.7216 | 0.3025 | 0.3458 | 0.3411 | 0.3607 | 0.4467 | 0.03 | - |
| | PredRNN | **0.7238** | 0.2708 | 0.3553 | 0.3702 | 0.4153 | 0.4621 | 0.15 | - |
| | SimVP | 0.7209 | 0.2793 | 0.3788 | 0.3803 | 0.4160 | 0.4920 | 0.01 | - |
| | Earthformer | 0.7102 | 0.3254 | 0.3556 | 0.3533 | 0.3838 | 0.4611 | 0.02 | - |
| | LDCast | 0.5772 | 0.2906 | 0.2193 | 0.2898 | 0.4598 | 0.2995 | 4.26 | 50 |
| | PreDiff | 0.6279 | 0.2217 | 0.3276 | 0.4271 | 0.6096 | 0.4498 | 73.09 | 1000 |
| | DiffCast | 0.6979 | 0.1948 | 0.3580 | 0.4555 | **0.6281** | 0.4751 | 5.20 | 250 |
| | STLDM | 0.7183 | **0.1929** | **0.3804** | **0.4662** | 0.6178 | **0.5024** | 0.51 | 20 |
| HKO-7 | ConvLSTM | 0.5987 | 0.3184 | 0.2905 | 0.2628 | 0.2774 | 0.4076 | 0.03 | - |
| | PredRNN | 0.5785 | 0.3131 | 0.2857 | 0.2872 | 0.3263 | 0.4026 | 0.15 | - |
| | SimVP | 0.6039 | 0.3596 | 0.3020 | 0.2852 | 0.3115 | 0.4236 | 0.02 | - |
| | Earthformer | 0.5864 | 0.3373 | 0.2817 | 0.2532 | 0.2704 | 0.3939 | 0.02 | - |
| | LDCast | 0.6003 | 0.2322 | 0.2145 | 0.3122 | 0.5345 | 0.3165 | 4.75 | 50 |
| | PreDiff | 0.5922 | 0.2391 | 0.2799 | 0.3787 | 0.5081 | 0.3973 | 70.80 | 1000 |
| | DiffCast | 0.6198 | 0.1949 | 0.3013 | 0.4084 | 0.6084 | 0.4240 | 20.50 | 250 |
| | STLDM | **0.6433** | **0.1943** | **0.3191** | **0.4413** | **0.6511** | **0.4447** | 0.55 | 20 |
| MeteoNet | ConvLSTM | 0.7938 | 0.2203 | 0.3619 | 0.3687 | 0.4130 | 0.5056 | 0.02 | - |
| | PredRNN | 0.8158 | 0.1419 | 0.3455 | 0.4904 | 0.5837 | 0.4810 | 0.16 | - |
| | SimVP | 0.8134 | 0.1734 | **0.3858** | 0.4467 | 0.5746 | **0.5358** | 0.02 | - |
| | Earthformer | 0.7806 | 0.2739 | 0.3401 | 0.3244 | 0.3488 | 0.4786 | 0.02 | - |
| | LDCast | 0.7654 | 0.1691 | 0.2620 | 0.3658 | 0.5685 | 0.3904 | 4.76 | 50 |
| | PreDiff | 0.7059 | 0.1543 | 0.2657 | 0.3854 | 0.5692 | 0.3782 | 70.80 | 1000 |
| | DiffCast | **0.8167** | 0.1280 | 0.3831 | 0.4771 | 0.6335 | 0.5328 | 21.30 | 250 |
| | STLDM | 0.8053 | **0.1275** | 0.3748 | **0.4921** | **0.6575** | 0.5233 | 0.51 | 20 |

tolerance, are measured as well. Besides that, HSS is a skill score that assesses the model's ability to predict precipitation events after considering the actual distribution of corresponding thresholds as mentioned above.

In addition, we also report two perceptual-related metrics: Structural Similarity Index Measure (SSIM) and Learned Perceptual Image Patch Similarity (LPIPS). SSIM is to evaluate the model's prediction in terms of structural similarity at the pixel level, while LPIPS is to assess the visual quality of predictions by measuring the distance between each pair of prediction and observation encoded by a pre-trained model.

To judge the model efficiency during the inference, we report the prediction time per sample, $T_{\text{sample}}$, on a single RTX3090 GPU. We also report the required sampling steps (aka denoising steps), $N$ for those diffusion-based models, i.e., LDCast, PreDiff, DiffCast, and STLDM.

## 4.2 Compared to the State-of-the-Art

To evaluate the performance of our STLDM, we report three diffusion-based probabilistic models: LDCast (Leinonen et al., 2023), PreDiff (Gao et al., 2023), and DiffCast (Yu et al., 2024) as baselines. Both LDCast and PreDiff perform the diffusion process in the latent space, while DiffCast is composed of a deterministic model and a diffusion model running in the pixel space. For fairness, the performance of these probabilistic models, including STLDM, is evaluated among 10 ensemble predictions. Additionally, various deterministic models: ConvLSTM (Shi et al., 2015), PredRNN (Wang et al., 2017), SimVP (Gao et al., 2022a) and Earthformer (Gao et al., 2022b) are included as references.

In Table 1, it is noted that STLDM is capable of achieving the best performance for most evaluation metrics, especially on the HKO-7 dataset. Our STLDM achieves the best performance in terms of LPIPS, which

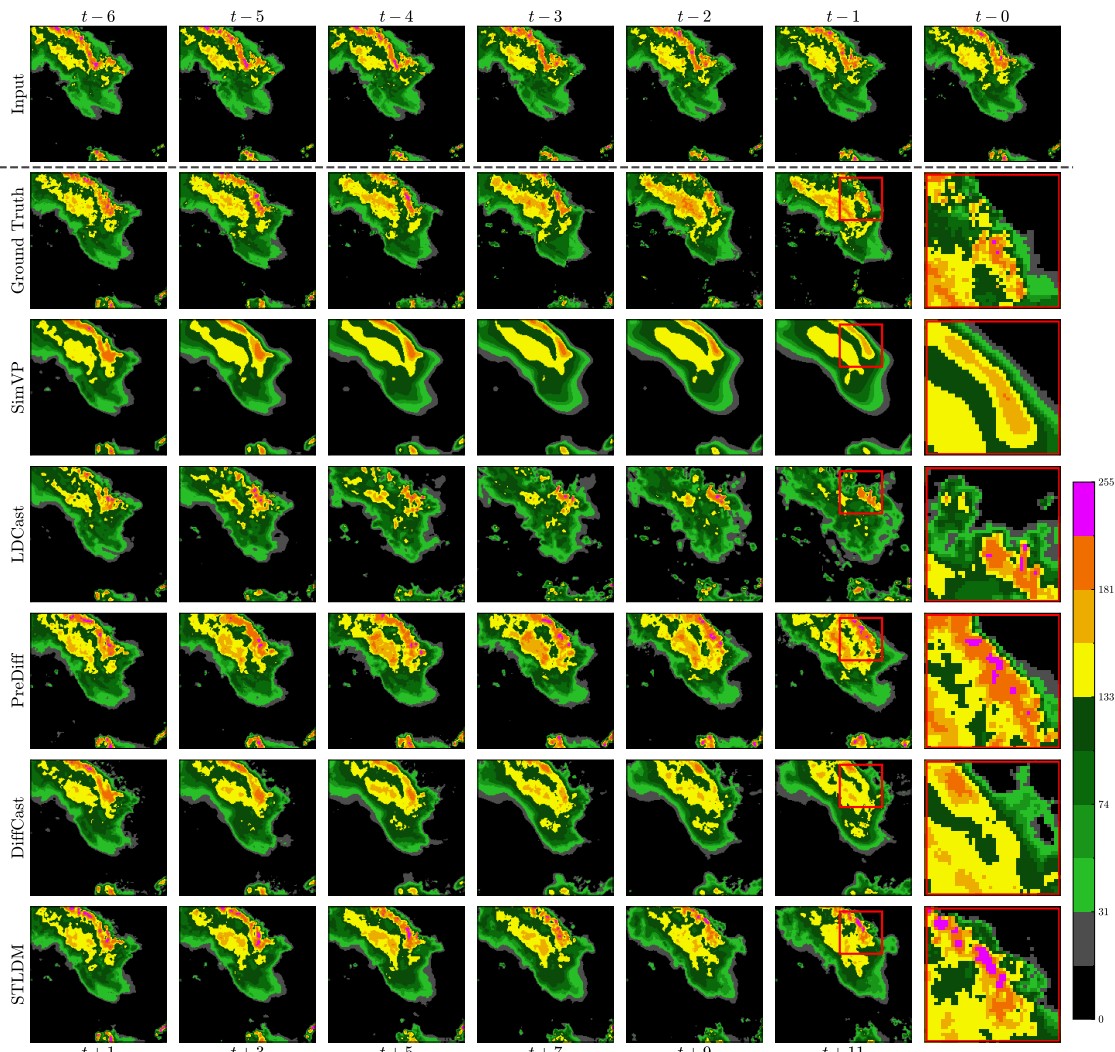

Figure 3: A set of sample predictions on the SEVIR test set. From top to bottom: Input, Ground truth, SimVP, PreDiff, DiffCast, and STLDM. The red region of the last prediction frame is zoomed in for a clearer comparison.

is a deep-learning-based perceptual metric. This is supported by the visualization shown in Figure 3 that STLDM has the prediction that is the perceptually closest to the ground truth. However, STLDM does not consistently outperform all baselines in every metric across the other two benchmarks. On the SEVIR dataset, STLLDM is 1.64% worse on $CSI_{16}$-m compared to DiffCast, while outperforming it on all other metrics. On the MeteoNet benchmark, STLDM shows a performance drop of 1.40% to 2.16% in SSIM, CSI-m, and HSS, but gains 0.39% to 3.79% improvements on the remaining metrics. Detailed visualizations on different datasets are shown in Figure 13, 14, and 15 respectively.

Since we adopted the latent-diffusion design as well as the model architecture, STLDM requires fewer sampling steps, enabling it to be 10X faster than DiffCast on the SEVIR dataset and 40X faster on both the HKO-7 and MeteoNet datasets. Notably, for models like DiffCast, the inference time scales exponentially with the image resolution. Hence, this efficiency gap between DiffCast and STLDM is even larger for data with higher resolution, as reported in Table 6. Moreover, those latent-diffusion related works: LDCast, PreDiff, and our STLDM have a relatively consistent sampling time across benchmarks, and STLDM is still the most efficient among them.

### 4.3 Analysis and Ablation Study

To identify which component contributes the most to STLDM's performance, we conducted several ablation studies, including the significance of the proposed loss term, $\mathcal{L}_C$, and different training strategies on every component of STLDM. Furthermore, we explore an alternative setup that treats the visual enhancement for every frame independently, rather than our current setting – Spatio-Temporal Visual Enhancement Task. Same as the evaluation settings above, all ablation studies are conducted here with ten ensemble predictions.

#### 4.3.1 Significance of Proposed Constraint Loss Term, $\mathcal{L}_C$

Table 2: Ablation study of the impact of $\mathcal{L}_C$ on the performace of STLDM with the SEVIR dataset. A better score is highlighted in **bold**.

| Existence of $\mathcal{L}_C$ | Metrics | | | | | |
| | SSIM↑ | LPIPS↓ | CSI-m↑ | CSI$_4$-m↑ | CSI$_{16}$-m↑ | HSS↑ |
| --- | --- | --- | --- | --- | --- | --- |
| ✗ | **0.7244** | 0.2046 | 0.3569 | 0.4533 | 0.6030 | 0.4659 |
| ✓ | 0.7183 | **0.1929** | **0.3804** | **0.4662** | **0.6178** | **0.5024** |

As discussed in Section 3.2.2, $\mathcal{L}_C$ is to constrain the final prediction, $\hat{Y}_{1:N}$, such that it has the global motion trend same as the first estimation, $\overline{Y}_{1:N}$. With the absence of $\mathcal{L}_C$, the conditioning network, $\Psi_\theta$, is trained only with the KL-divergence and diffusion loss in Terms C and E respectively. Both terms implicitly constrain the prediction of $\Psi_\theta$, resulting in a not well-constrained $\overline{Y}_{1:N}$ as shown in Figure 7, i.e., a noisy prediction. This noisy $\overline{Y}_{1:N}$ does not help much the final prediction, $\hat{Y}_{1:N}$, by the latent denoising network, $D_\theta$, ultimately causing STLDM to fail to predict the precipitation events accurately.

The claim above is supported by the comparison of the model performance with and without $\mathcal{L}_C$ reported in Table 2. From Table 2, these two cases have similar performance in both SSIM and LPIPS, corresponding to the visual assessment of prediction. The existence of $\mathcal{L}_C$ during training improves all forecasting skill scores: both CSI and HSS notably. This observation is confirmed by Figure 7 that $\hat{Y}_{1:N}$ tends to over-predict due to its noisy first estimation, $\overline{Y}_{1:N}$ from $\Psi_\theta$ in the absence of $\mathcal{L}_C$ during training. In contrast, $\mathcal{L}_C$ regularizes the first estimation, $\overline{Y}_{1:N}$, thereby enabling more accurate final predictions. This indicates that $\mathcal{L}_C$ plays a crucial role in improving STLDM's accuracy via constraining the first estimation of $\Psi_\theta$.

Consequently, explicitly constraining the conditioning network, $\Psi_\theta$, with the proposed constraint loss, $\mathcal{L}_C$, consistently improves the performance of STLDM as it ensures the correctness of its prediction.

#### 4.3.2 Different Training Strategies

Table 3: Ablation study on different training strategies: Which model components are trained along with the denoising network on the SEVIR dataset. A better score is highlighted in **bold**.

| Strategy | Components Trained Together | | | Metrics | | | | | |
| | $\{\mathcal{E}, \mathcal{D}\}$ | $\Psi_\theta$ | $D_\theta$ | SSIM↑ | LPIPS↓ | CSI-m↑ | CSI$_4$-m↑ | CSI$_{16}$-m↑ | HSS↑ |
| --- | --- | --- | --- | --- | --- | --- | --- | --- | --- |
| A | ✗ | ✗ | ✓ | 0.7086 | 0.2121 | 0.3809 | 0.4676 | 0.6209 | 0.5028 |
| B | ✗ | ✓ | ✓ | 0.7173 | 0.1955 | **0.3822** | **0.4680** | **0.6209** | **0.5043** |
| C | ✓ | ✓ | ✓ | **0.7183** | **0.1929** | 0.3804 | 0.4662 | 0.6178 | 0.5024 |

In this part, we investigate the impact of different training strategies on the performance of STLDM. Beyond our current end-to-end tuning strategy (Strategy C), we also report two alternatives: (i) Train every component individually (Strategy A), and (ii) Jointly train both the Latent Denoising Network, $D_\theta$, and Conditioning Network, $\Psi_\theta$, without further tuning the pre-trained Variational Autoencoder (Strategy B).

These two approaches share a common feature: they do not require further tuning of pre-trained VAE during the training of the denoising network, $D_\theta$, resulting in a lower GPU memory demand for model training. Hence, these strategies are widely implemented because of their training efficiency. Additionally, using a pre-trained Conditioning Network further decreases the demand for GPU memory and also benefits from exposure to a broader pre-training dataset, potentially improving generalization.

From Table 3, all these three approaches have comparable performance in all forecasting skill scores, with the case that Strategy B: Tuning denoising network, $D_\theta$, along with the Conditioning Network, $\Psi_\theta$, has a slightly better performance. Furthermore, it is noted that our current training strategy, i.e., Strategy C: End-to-End Tuning, yields the best performance in terms of perceptual metrics, specifically SSIM and LPIPS, while Strategy A: Tuning every component individually achieves the worst performance in these metrics.

Moreover, we study the training convergence of these approaches with the validation MSE and LPIPS reported in Figure 6. From the plots in Figure 6, Strategy A with both pre-trained VAE and $\Psi_\theta$ has the fastest convergence speed but fails to achieve those low metrics as the other approaches could. Our current training setting, i.e., Strategy C, exhibits slightly better validation performance than Strategy B after training. This reveals that the collaboration among all components in STLDM is important in resulting in better model performance, especially the cooperation between the Conditioning Network, $\Psi_\theta$, and the Latent Denoising Network, $D_\theta$.

### 4.3.3 Can We Enhance Every Frames Independently?

Table 4: Ablation study on different kinds of visual enhancement tasks on the HKO-7 dataset. A better score is highlighted in **bold**.

| Types of Enhancement | LPIPS↓ | FVD↓ | $T_{\text{sample}}$ |
|---|---|---|---|
| Spatial | **0.1878** | 241.17 | 0.4157 |
| Spatio-Temporal | 0.1943 | **87.14** | 0.5533 |

As mentioned in Section 3.2.1, we reformulate the nowcasting task as two sub-tasks: Forecasting and Enhancement in sequence. In our current setting, we treat this visual enhancement task as a spatio-temporal task with a Spatial Temporal Latent Denoising Network, $D_\theta$. Here, we explore an alternative formulation that treats enhancement as a purely spatial task by removing all Temporal Attention in $D_\theta$.

In addition to the required inference time, $T_{\text{sample}}$, on a single RTX3090 GPU, we report two metrics: LPIPS (Zhang et al., 2018) and FVD (Unterthiner et al., 2019) for assessing the perceptual scores in spatial and spatio-temporal, respectively, on the HKO-7 dataset in Table 4 as well. From Table 4, we observe that treating this visual enhancement task frame-wisely has a slightly better LPIPS score and shorter inference time, $T_{\text{sample}}$. However, STLDM's spatio-temporal enhancement task achieves a significantly better FVD score, indicating superior temporal consistency. This finding is further supported by the visualizations in Figure 9 and 10 that the cloud motion in red bounding boxes in the settings of the Spatial Visual Enhancement task has a drastic change, resulting in the temporal inconsistency; while our current setting and the ground truth share a similar cloud movement, i.e., steady and slow. These results highlight the importance of treating this enhancement task as a spatiotemporal task to achieve better temporal coherence.

### 4.3.4 Application in Meteorological Operation

Despite their fast inference time, low resource requirement, and reasonable accuracy, the radar nowcasts predicted by deterministic deep learning models are inadequate for supporting the prediction of convective weather systems, as they commonly struggle with blurriness issues, as shown in Figure 1. In assessing the evolution of significant convective weather systems, it is critical to preserve both the spatial structure of high-intensity reflectivity pixels and their temporal consistency. STLDM is capable of improving the prediction of small-scale and complex structures of significant convective systems over a longer time range, resulting in sharp and realistic nowcasts. This enables the weather forecasters to issue more timely alerts and warnings to the public. Furthermore, STLDM's fast inference time relative to other diffusion models enables probabilistic nowcasting with a large ensemble size in real-time forecasting operations. This facilitates the assessment of the likelihood and alternative scenarios regarding the movement and severity of high-impact rainstorm events, thereby mitigating potential societal losses of life and property.

## 5 Conclusion and Future Work

In this work, we propose a simple yet effective Spatio-Temporal Latent Diffusion Model (STLDM) for precipitation nowcasting, based on the idea of reformulating this task into two sub-tasks in sequence: Forecasting and Enhancement. STLDM is composed of three modules: a Variational AutoEncoder, a Conditioning Network/Translator, and a Latent Denoising Network. Extensive experimental results across multiple radar datasets demonstrate that STLDM outperforms existing techniques, validating its effectiveness. Furthermore, we argue that introducing conditional regularization on the Translator during training consistently improves the model's performance. Moreover, we also reveal that training the Latent Denoising Network along with other components yields better results than training each component individually. Lastly, we also emphasize that modeling the visual enhancement task as a spatio-temporal task ensures better temporal consistency.

**Limitations and future work.** Despite STLDM's competitive performance, it still struggles to accurately forecast precipitation events that originate outside the observation region or are driven by unknown factors, as illustrated in Figure 16. A possible solution is to introduce a multimodal model capable of capturing precipitation events beyond the observation window. Additionally, STLDM must be retrained for a new dataset in a different region. Although the precipitation events in different regions exhibit unique characteristics, they also share underlying common features, such as the governing physical principles. Therefore, our future direction is to develop a unified and physics-informed multimodal model across multiple benchmarks, offering not only broader generalization but also improved physical interpretability.

### Acknowledgments

This work has been made possible by a Research Impact Fund project (RIF R6003-21) and a General Research Fund project (GRF 16203224) funded by the Research Grants Council (RGC) of the Hong Kong Government.

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

# A Model Architecture of STLDM

In this section, we specify the details of every component of STLDM: a Variational autoencoder, a conditioning network, and a latent denoising network. Additionally, STLDM's hyperparameters for both the training and inference processes are reported in this section.

## A.1 Variational AutoEncoder, $\{\mathcal{E}, \mathcal{D}\}$

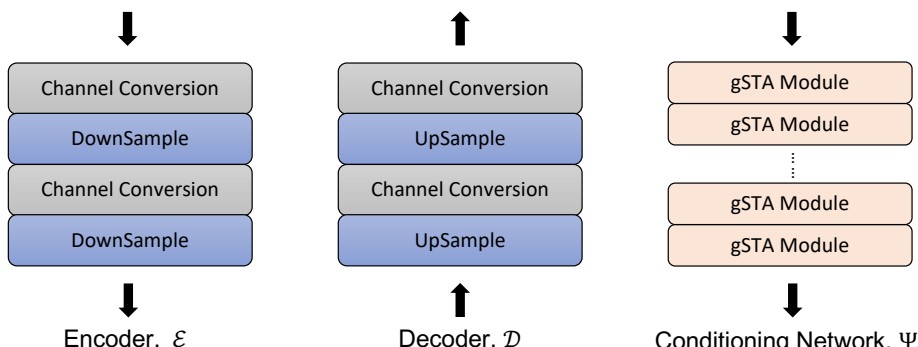

Figure 4: Illustration of the implemented encoder, $\mathcal{E}$, decoder, $\mathcal{D}$ and conditioning network, $\Psi_\theta$.

We follow these previous works (Gao et al., 2022a; Tan et al., 2023b) to build a Variational autoencoder, $\{\mathcal{E}, \mathcal{D}\}$, as shown in Figure 4 that solely relies on convolutional operations. The only change we made is the removal of the skip connection between the encoder and the decoder. Specifically, the input radar frames at time $t$, $x_t \in \mathbb{R}^{1 \times 128 \times 128}$, are encoded to the corresponding latent variables, $z_t \in \mathbb{R}^{32 \times 32 \times 32}$, by the encoder while the decoder decodes the predicted latent variables back into the predicted frames.

The components inside the encoder and decoder are described here:

- **Channel Conversion**: $3 \times 3$ convolution layer, Group Normalization over groups of 2 followed by a leaky ReLU activation layer.

- **Down/Upsample**: Transposed convolutional operations that downsample or upsample by a spatial factor of 2.

## A.2 Conditioning Network/Translator, $\Psi_\theta$

We stack several Gated Spatio-Temporal Attention (gSTA) modules (Tan et al., 2023b) as the conditioning network, $\Psi_\theta$, for modeling the underlying relation between the encoded input frames and the target output frames as illustrated in Figure 4. The gSTA module is also solely composed of convolutional operations for modeling spatio-temporal features, imitating the spatio-temporal attention mechanism with a large kernel convolution. Every gSTA module consists of a depth-wise convolution, a depth-wise dilation convolution, and a channel-wise convolution as illustrated in Figure 5.

For constraining the conditional vector on thw Latent Denoising Network (a.k.a the prediction from $\Psi_\theta$), we regularize the decoded prediction from $\Psi_\theta$, $\overline{Y}_{1:N}$, with the ground truth as stated in the loss term, $\mathcal{L}_C$, included in Equation 10.

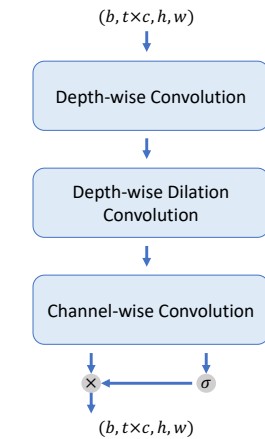

Figure 5: Gated Spatio-Temporal Attention (gSTA) module

## A.3 Latent Denoising Network, $D_\theta$

The architecture of the latent denoising network, $D_\theta$, included in our proposed STLDM is shown in Figure 2. In this section, we elaborate on different components inside $D_\theta$:

- **ResBlock**: It is composed of two subblocks and followed by a $1 \times 1$ convolution for channel conversion. Each subblock inside contains a $3 \times 3$ convolution, group normalization over groups of 8, and a SiLU activation.

- **Down/Upsample**: These are the convolutional operations (with a kernel size of 4, padding of 1, and stride of 2) that downsamples or upsamples by a spatial factor of 2.

- **Spatial Attention**: A self-attention between each pixel in every layer after interpreting the input as batches of independent frames (by shifting the temporal axis into batch dimension), i.e., $[b, t, c, h, w] \rightarrow [b \times t, c, h, w]$.

- **Linearized Spatial Attention (L-Spatial)**: This linearized attention is a self-attention between each pixel within a patch with patch size, $p$. This is achieved by considering every independent patch as an attention head as well, after patching the query, key, and value. Patch size, $p$, is halved for each Downsampling Block while $p$ is doubled for each Upsampling Block. The order of computing the self-attention follows Equation 15.

- **Temporal Attention**: This is a self-attention between each frame in every layer along the temporal dimension with the following reshape: $[b, t, c, h, w] \rightarrow [b \times h \times w, c, t]$.

where $b$, $t$, $c$, $h$, and $w$ are denoted as batch, time, channel, height, and width, respectively.

### A.4  Hyper-Parameters of Training and Inference

We trained the models for 200k training steps in total on all benchmarks with a batch size of 4. The learning rate is scheduled with a 2k steps warm-up period, followed by a Cosine Annealing Scheduler decaying from the peak learning rate of 1e−4. Besides that, we set the total sampling steps of STLDM to 50.

During the inference process, we employ the DDIM technique (Song et al., 2021) of 20 sampling steps and the Classifier-Free Guidance (Ho & Salimans, 2022) with the strength of 1.0.

## B  Details about Task Reformulation

In this section, we provide the details of the task reformulation process. Recall that, the objective of this nowcasting task is to find a predicted sequences, $\hat{Y}_{1:N}$, which has the highest conditional probability, $p(\hat{Y}_{1:N}|X_{1:M})$, with the given input radar sequences, $X_{1:M}$.

First, the conditional probability of both events $A$ and $B$ happening given that event $C$ occurred could be rewritten as the product of two conditional probabilities:

$$
\begin{aligned}
P(A, B|C) &= \frac{P(A, B, C)}{P(C)} \\
&= \frac{P(A|B, C)P(B, C)}{P(C)} \\
&= P(A|B, C)P(B|C)
\end{aligned}
$$

Motivated by the idea above, we introduce an intermediate variable (aka the first estimation), $\overline{Y}_{1:N}$, then we could rewrite the conditional probability of this task objective as follows:

$$
\begin{aligned}
p(\hat{Y}_{1:N}|X_{1:M}) &= \int p(\hat{Y}_{1:N}, \overline{Y}_{1:N}|X_{1:M}) d\overline{Y}_{1:N} \\
&= \int p(\hat{Y}_{1:N}|X_{1:M}, \overline{Y}_{1:N}) p(\overline{Y}_{1:N}|X_{1:M}) d\overline{Y}_{1:N}
\end{aligned}
$$

Hence, we introduce an additional task objective – **Forecasting** that constraining the first estimation, $\overline{Y}_{1:N}$, to this nowcasting task:

$$p(\hat{Y}_{1:N}, \overline{Y}_{1:N}|X_{1:M}) = p(\hat{Y}_{1:N}|\overline{Y}_{1:N}, X_{1:M})p(\overline{Y}_{1:N}|X_{1:M})$$

$$\nabla_\theta \log p(\hat{Y}_{1:N}, \overline{Y}_{1:N}|X_{1:M}) = \underbrace{\nabla_\theta \log p(\overline{Y}_{1:N}|X_{1:M})}_{\text{Forecasting}} + \underbrace{\nabla_\theta \log p(\hat{Y}_{1:N}|\overline{Y}_{1:N}, X_{1:M})}_{\text{Visual Enhancement}},$$

where $\theta$ refers to the model parameters. The first term is obligated for the forecasting objective, while the latter term is responsible for the visual enhancement goal.

To fulfill this, we reformulate the objective of the nowcasting task into two sequential sub-tasks: Forecasting and Enhancement, with proposing a constraint loss, $\mathcal{L}_C$, mentioned in Equation 10.

## C  Validation MSE and LPIPS of Different Training Strategies

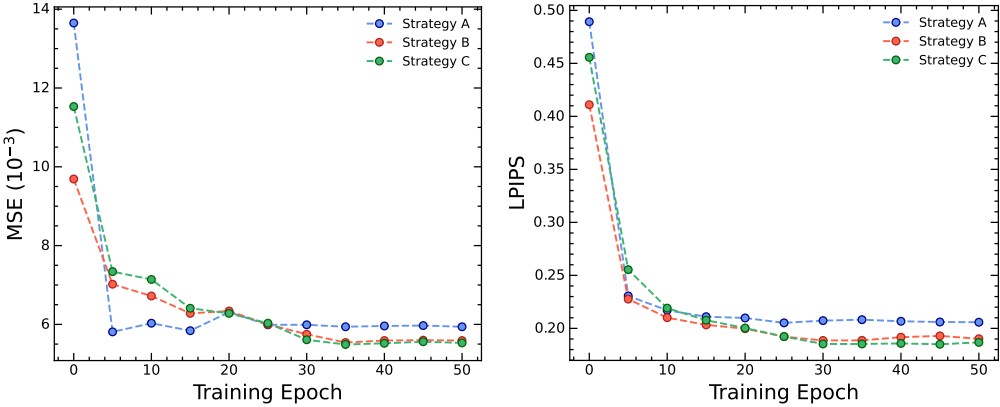

Figure 6: Validation MSE and LPIPS of different training strategies of STLDM during the training process.

## D  Evaluation on a High-Resolution Dataset

To evaluate the feasibility of STLDM at different spatial resolutions, we also report its quantitative results on the HKO-7 dataset with a spatial size of $256 \times 256$ with two baselines: LDCast and DiffCast in Table 5. The forecasting task remains the same – predicting the next 20 frames given 5 input frames.

From Table 5, we observe that STLDM achieves the best performance across all metrics. This result is consistent with its performance on the downscaled HKO-7 dataset reported in Table 1. This makes us conclude that STLDM consistently outperforms the baselines on the HKO-7 dataset regardless of spatial resolution. A detailed visualization is shown in Figure 11.

Table 5: Performance comparison of our proposed STLDM with two baselines: LDCast and DiffCast on the HKO-7 dataset with the spatial size of 256. The best score among all models is highlighted in **bold.**

| Model | Metrics | | | | | |
|---|---|---|---|---|---|---|
| | SSIM↑ | LPIPS↓ | CSI-m↑ | CSI$_4$-m↑ | CSI$_{16}$-m↑ | HSS↑ |
| LDCast | 0.6776 | 0.2476 | 0.1558 | 0.2096 | 0.3675 | 0.2375 |
| DiffCast | 0.6986 | 0.1854 | 0.3111 | 0.3799 | 0.5303 | 0.4372 |
| STLDM | **0.7097** | **0.1840** | **0.3778** | **0.4137** | **0.5665** | **0.4671** |

# E   Running Time of STLDM and Baselines

We have demonstrated that STLDM is the most effective among diffusion-based models, achieving state-of-the-art performance across multiple real-life radar echo datasets. In this section, we analyze STLDM's efficiency from two perspectives: inference time and training cost.

To compare inference efficiency with other baselines, we report their inference times on the HKO-7 benchmark at three spatial resolutions: 128, 256, and 512 in Table 6. From Table 6, we concluded that STLDM consistently achieves the fastest sampling speed at every spatial scale, resulting in the shortest inference time. Moreover, the performance gap between STLDM and the baselines is increasing with the spatial scale.

In addition, we also reported the average training time of these models on the HKO-7 benchmark with a spatial size of 128 and a batch size of 4 over 10k training steps in Table 7. From Table 7, despite being trained from end to end, STLDM still requires the least training time, leading to the lowest training cost. Table 6 and 7 highlight that STLDM is the most efficient model, achieving both the shortest inference time and the lowest training cost.

Table 6: Inference time (in seconds) of each model on the HKO-7 dataset with different spatial sizes, using a single RTX3090 GPU.

| Model | Sampling Steps | Inference Time (s) | | |
|---|---|---|---|---|
| | | 128 | 256 | 512 |
| LDCast | 50 | 4.76 | 12.17 | 46.69 |
| PreDiff | 1000 | 70.80 | 222.73 | 900.15 |
| DiffCast | 250 | 20.50 | 25.06 | 99.90 |
| STLDM | 20 | 0.55 | 0.66 | 2.41 |

Table 7: Average training time for each model on the HKO-7 benchmark with a batch of 4 over 10k training steps, using a single RTX3090 GPU.

| Model | Training Time |
|---|---|
| LDCast with a pre-trained VAE | 3 hours 48 minutes |
| DiffCast | 4 hours 12 minutes |
| STLDM from end to end | 2 hours 23 minutes |

# F   The Implementation of Classifier-Free Guidance and Its Significance

As discussed in Section 3.1.2, the technique of Classifier-Free Guidance (CFG) Ho & Salimans (2022) is to ensure that the generated sample is constrained with the given conditional variable, $c$, which is the first estimation, $\bar{z}_{1:N}$, in the proposed STLDM. In this section, we describe how CFG is implemented in STLDM and study the impact of CFG on the performance of STLDM.

As mentioned in Section 3.1.2, CFG involves two cases: both the conditional and unconditional cases. Following prior works, we adopt a joint training strategy where the conditioning signal is randomly dropped with the probability of 15% (i.e, 85% conditional and 15% unconditional training). During inference, at each denoising timestep, we compute both the conditional and unconditional noise predictions, which is two forward passes per step. We then combine these predictions and modify the denoising score according to Equation 7 with guidance strength, $w = 1.0$. This ensures that CFG is applied throughout the sampling process.

Specifically, to apply CFG in our STLDM, we concatenate the latent output from the Translator module, $\bar{z}_{1:N}$, with the noisy latent input, $z_{1:N}^t$, along the channel dimension Srivastava et al. (2024) at the first ResBlock of each Downsampling Block, as illustrated in Figure 2. This corresponds to the conditional case. For the unconditional case, we replace $\bar{z}_{1:N}$ with a zero tensor of the same size, which serves as the null embedding here.

To validate the effectiveness of CFG, we conducted an ablation study on the SEVIR dataset here. As reported in Table 8, applying the CFG technique consistently improves the performance of STLDM across all evaluation metrics. This improvement is validated by the visualization shown in Figure 12. These results demonstrate that incorporating CFG provides consistent performance improvements, indicating its effectiveness.

Table 8: Ablation study of the impact of CFG on the performance of STLDM with the SEVIR dataset. A better score is highlighted in **bold**.

| Existence of CFG | Metrics | | | | | |
|---|---|---|---|---|---|---|
| | SSIM↑ | LPIPS↓ | CSI-m↑ | $CSI_4$-m↑ | $CSI_{16}$-m↑ | HSS↑ |
| ✗ | 0.7041 | 0.2129 | 0.3719 | 0.4376 | 0.5614 | 0.4898 |
| ✓ | **0.7183** | **0.1929** | **0.3804** | **0.4662** | **0.6178** | **0.5024** |

## G   More Visualizations

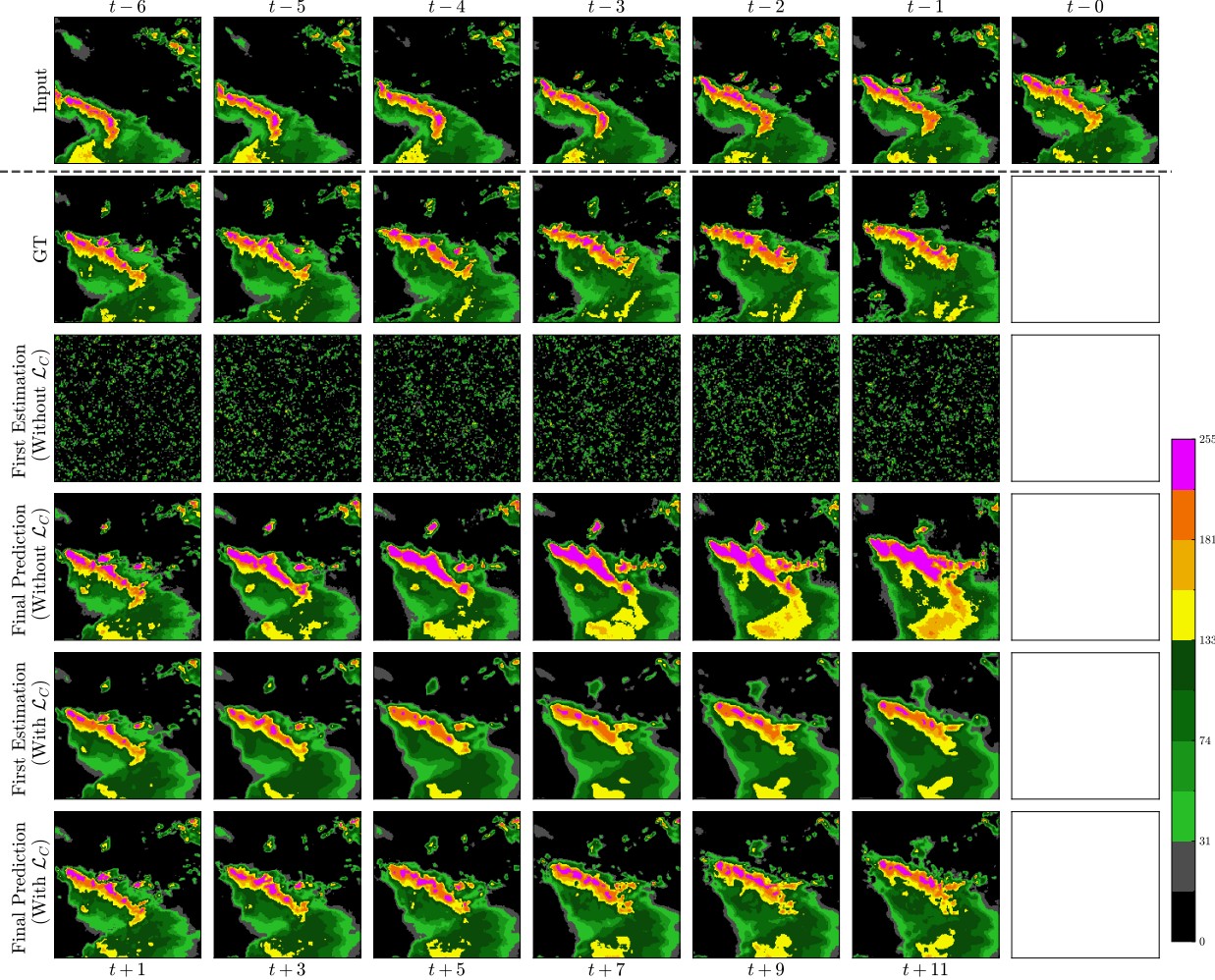

Figure 7: A set of first estimation and final prediction from STLDM trained in both cases, that with and without the proposed Constraint Loss, $\mathcal{L}_C$, as mentioned in Term A during training on the SEVIR test set.

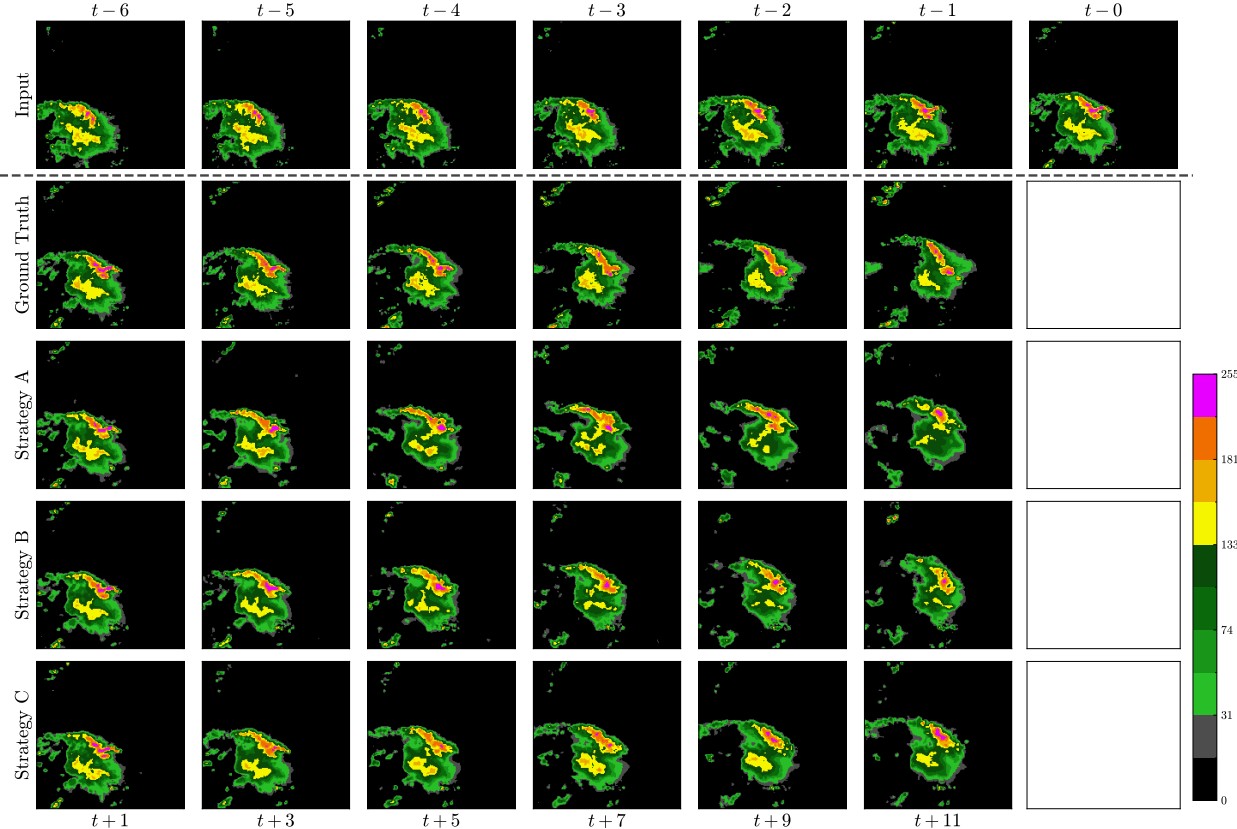

Figure 8: A set of sample predictions from STLDM trained with different training strategies as mentioned in Section 4.3.2 on the SEVIR test set.

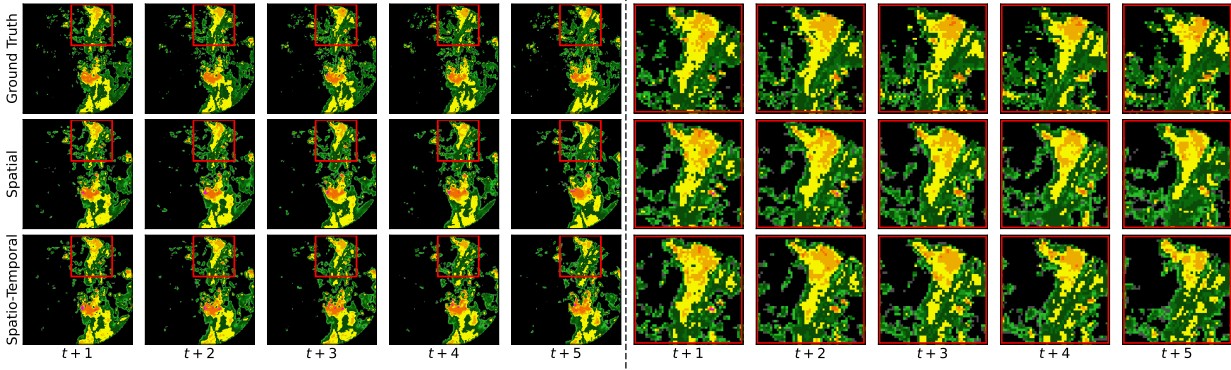

Figure 9: A set of sample predictions from STLDM with two different kinds of enhancement: Spatial and Spatio-Temporal on the HKO-7 test set. The red region of the first five frames is zoomed in for a clearer comparison.

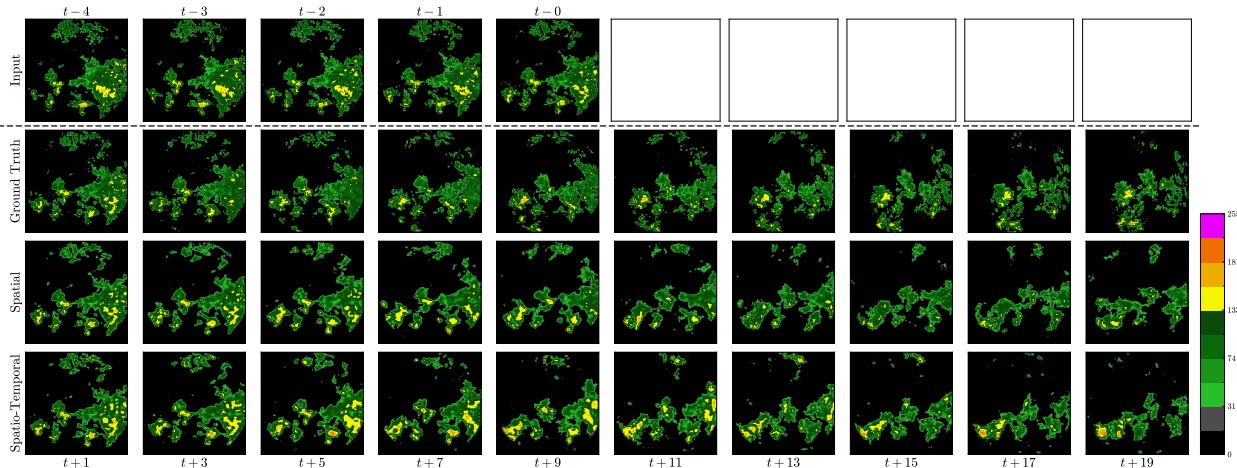

Figure 10: A set of sample predictions from STLDM with two different kinds of visual enhancement settings: Spatial and Spatio-Temporal, as mentioned in Section 4.3.3 on the HKO-7 test set.

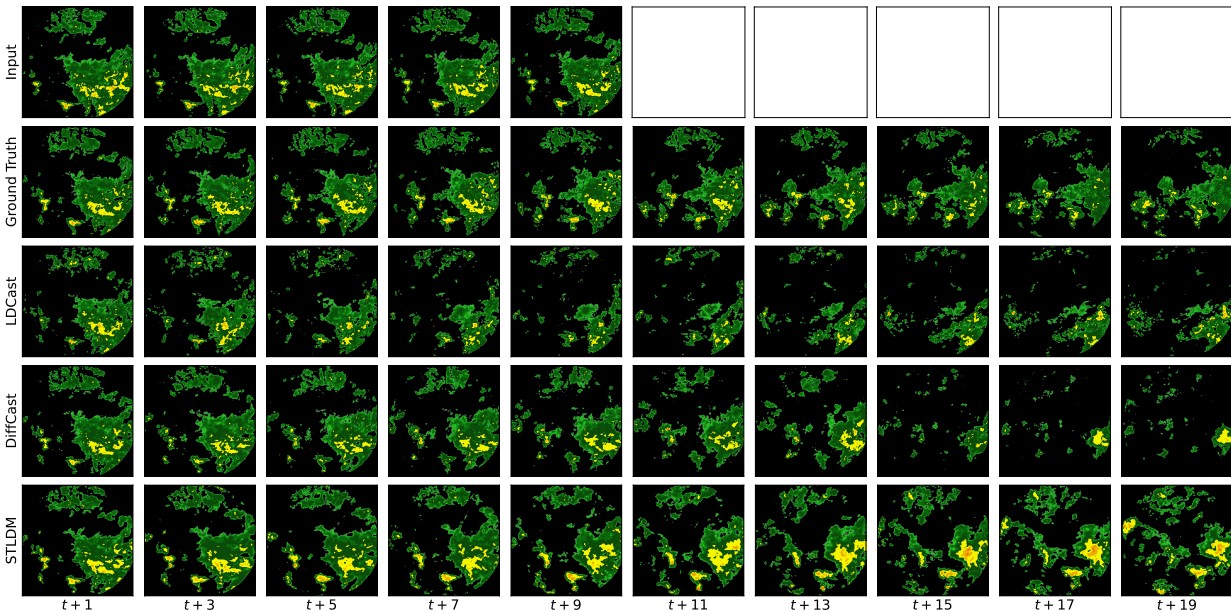

Figure 11: A set of sample predictions on the HKO-7 test set with a spatial size of 256. From top to bottom: Input, Ground truth, LDCast, DiffCast, and STLDM.

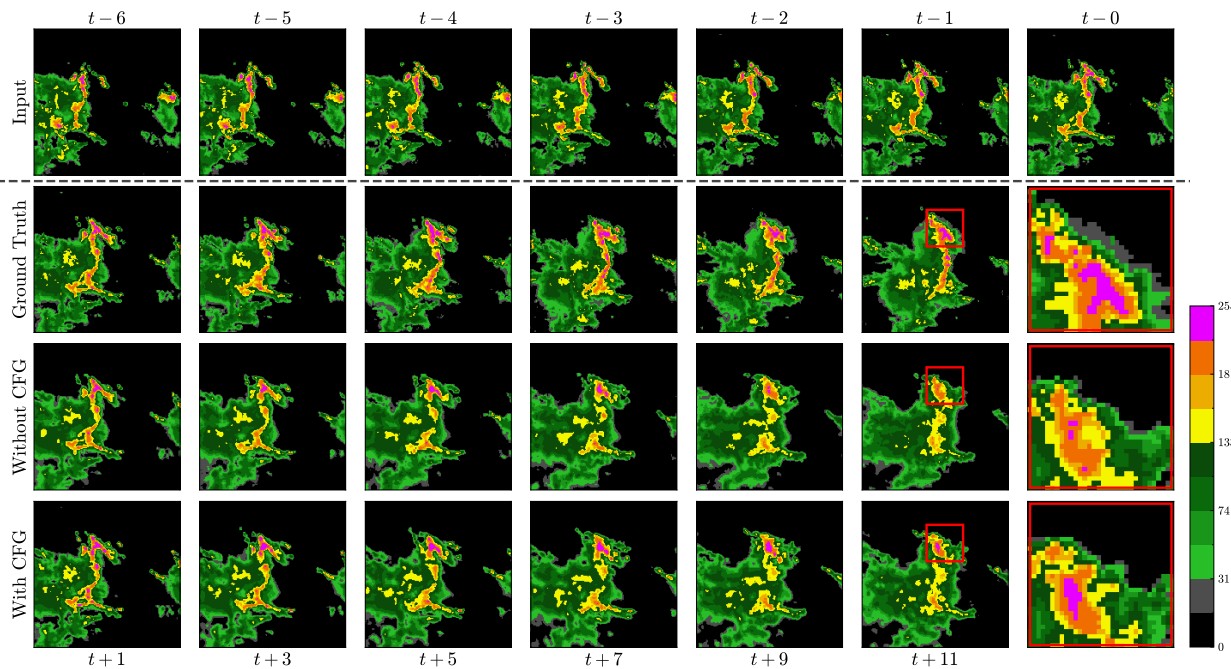

Figure 12: A set of sample predictions on the SEVIR test set with the absence and existence of Classifier-Free Guidance (CFG). The red region of the last prediction frame is zoomed in for a clearer comparison.

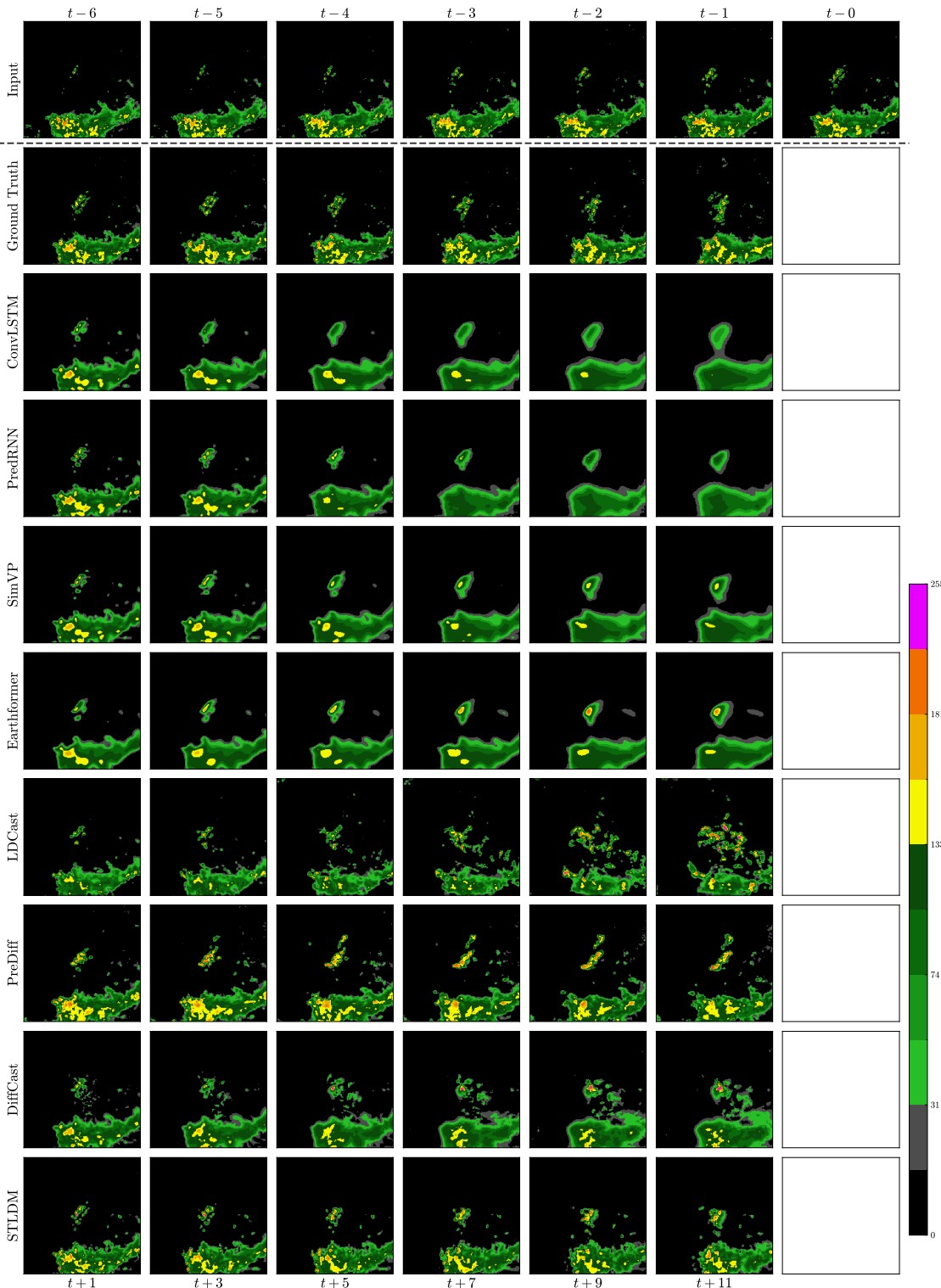

Figure 13: A set of sample predictions on the SEVIR test set. From top to bottom: Input, Ground truth, ConvLSTM, PredRNN, SimVP, Earthformer, LDCast, PreDiff, DiffCast, and STLDM.

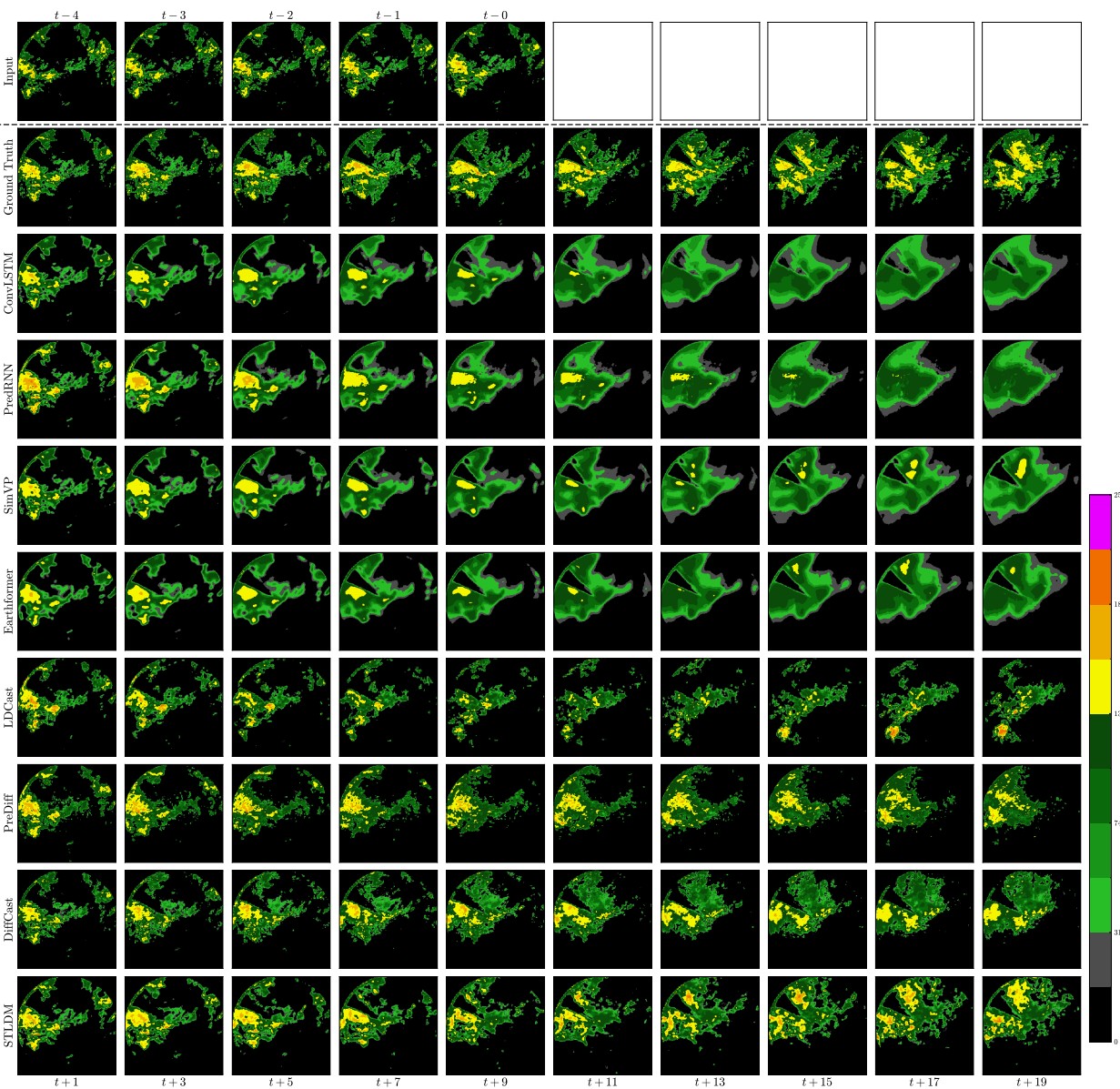

Figure 14: A set of sample predictions on the HKO-7 test set. From top to bottom: Input, Ground truth, ConvLSTM, PredRNN, SimVP, Earthformer, LDCast, PreDiff, DiffCast, and STLDM.

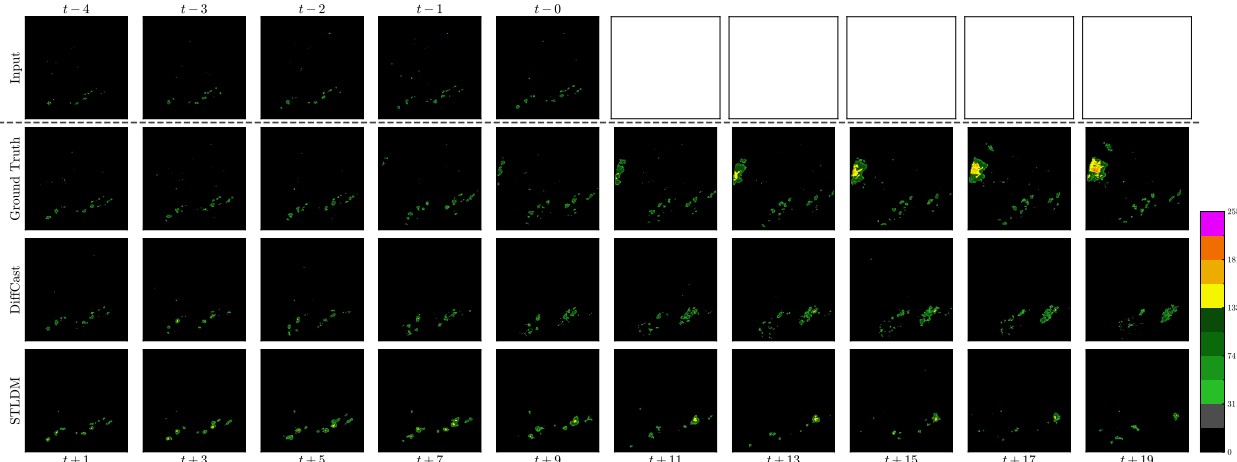

Figure 15: A set of sample predictions on the MeteoNet test set. From top to bottom: Input, Ground truth, ConvLSTM, PredRNN, SimVP, Earthformer, LDCast, DiffCast, and STLDM.

Figure 16: A set of sample predictions that both DiffCast and STLDM failed to predict the precipitation events spawning on the left due to the limited observation region on the HKO-7 test set.

