# OpenReview forum: "STLDM: Spatio-Temporal Latent Diffusion Model for Precipitation Nowcasting"
_TMLR — Accepted by TMLR_

### Review · Reviewer_LXNm · 2025-09-05

**Summary Of Contributions:**

Contributions of the manuscript:

- Reformulates precipitation nowcasting into two sequential subtasks: Forecasting (global motion trend) and Enhancement (fine-grained details). Proposes STLDM (Spatio-Temporal Latent Diffusion Model) combining a VAE, Conditioning Network, and Latent Denoising Network, trained end-to-end.
- Introduces a constraint loss (LC) to align global motion and enhance final predictions.
- Demonstrates state-of-the-art performance on multiple radar datasets (SEVIR, HKO-7, MeteoNet) across both accuracy and perceptual metrics.
- Achieves substantially faster inference than prior diffusion-based models due to latent-space design.

Strengths of the manuscript:
- Balanced trade-off: Improves both accuracy (like deterministic models) and sharpness/realism (like generative models).
- Efficiency: 10–40× faster inference than pixel-space diffusion models.
- Robustness: Strong performance across diverse datasets and metrics.
- Novelty: First to jointly train VAE, conditioning, and diffusion modules for nowcasting.
- Careful ablation studies support design choices (e.g., effectiveness of constraint loss).

Weaknesses of the manuscript:
- Inconsistent performance: Does not dominate across all datasets and metrics (e.g., CSI/SSIM on SEVIR and MeteoNet).
- Complex architecture: Although efficient in inference, training involves multiple interacting components (VAE + diffusion + translator), which may increase training cost and tuning difficulty.
- Limited interpretability: Like most deep generative models, lacks transparency for decision-making in meteorological operations.
- Evaluation scope: Tested mainly on downscaled radar datasets; scalability to higher resolution or operational deployment is not fully explored.

**Audience:**

Yes

**Audience Explanation:**

The TMLR audience includes researchers in machine learning and applications such as spatiotemporal modeling, generative models, and weather forecasting, all of which are directly addressed by this paper.

**Broader Impact Concerns:**

A reasonable concern here is that the Broader Impact Statement may not fully cover the societal and ethical implications of this work. While precipitation nowcasting is beneficial for disaster preparedness and resource planning, deploying a generative model for weather prediction also carries risks: inaccurate or overconfident forecasts could mislead policymakers or the public, with potential consequences for safety and resource allocation. Additionally, the complexity and lack of interpretability of diffusion models may reduce trust and hinder accountability in operational meteorology. The paper would benefit from explicitly acknowledging these issues, discussing the importance of uncertainty communication, and clarifying that such models should complement, not replace, established forecasting systems.

**Claims And Evidence:**

Yes

**Claims Explanation:**

Yes, the claims are generally supported by accurate and convincing evidence through extensive experiments on three benchmark datasets, use of multiple established metrics, and clear ablation studies that validate the proposed components. The efficiency and balanced performance claims are particularly well substantiated, with quantitative results and visualizations illustrating improvements. However, some claims of superiority are overstated, as the model does not consistently outperform baselines across all datasets and metrics, and evidence for operational scalability remains limited.

**Requested Changes:**

- Tone down claims of universal superiority; highlight mixed results across datasets/metrics.
- Expand discussion of limitations, especially scalability to higher-resolution or operational settings.
- Clarify training complexity and computational costs relative to baselines.
- Improve justification of how the constraint loss contributes to performance.
- Provide more discussion on interpretability and potential use in real-world meteorological operations, e.g. well defined road for real practice.

---

> ### Author Response · Authors · 2025-09-25
> **Response to Reviewer LXNm**
>
> We would like to thank the Reviewer LXNm's feedback and suggestions. We have modified our manuscript accordingly and highlighted those changes in blue. Here are our responses to each of Reviewer LXNm's concerns.
>
> **Weakness**
>
> > **W1**: Inconsistent performance
>
> We acknowledge that our proposed STLDM does not achieve dominant performance across all evaluation metrics and benchmarks - specifically, CSI$_{16}$-m on the SEVIR dataset and {SSIM, CSI-m, HSS} on the MeteoNet dataset. However, it is worth noting that STLDM achieves improvement ranging from 0.31% to 8.06%, while its performance degradation is limited to the range from 1.40 % to 2.17 %. This demonstrates that even in the cases that STLDM does not dominate, its shortcomings are marginal compared to the significant improvements it gains.
>
> >**W2**: Complex architecture
>
> Although STLDM integrates multiple components (VAE, Translator and Latent Denoising Network), this complexity does not result in a heavy training cost. To measure the training cost, we report the average time required to train STLDM and two baselines: LDCast and DiffCast, over 10k steps with a batch size of 4 on the HKO-7 dataset, using a Nvidia RTX3090 GPU. The results are summarised below:
>
> | Model | Training Time |
> |:------|:-----:|
> | LDCast with a pre-trained VAE| 3 hours 48 minutes |
> | DiffCast | 4 hours 12 minutes |
> | STLDM from end to end | 2 hours 23 minutes |
>
> As shown in the table above and Table 1 in the paper, STLDM is in fact more efficient in both training and inference, achieving shorter training time despite its richer architecture. This demonstrates that the design of STLDM is well-justified, as it leads to practical advantages without imposing extra training overhead.
>
> > **W3**: Limited interpretability
>
> We acknowledge that, like most deep generative models, STLDM operates as a black box and lacks full transparency and physical interpretability. However, its faster inference speed and ability to produce realistic, sharp predictions enable probabilistic nowcasting with a large ensemble size in real-time forecasting operations. This allows weather forecasters to issue more timely alerts to the public.
>
> > **W4**: Evaluation Scope
>
> Due to time limitations, we were only able to train two baselines – LD-Cast and DiffCast, together with our proposed STLDM on a higher resolution dataset, namely the HKO-7 dataset, with an image size of 256x256 under the same task setting (i.e., 5in-20out).  Their performance is reported in the table below:
>
> | Model | SSIM | LPIPS | CSI-m | CSI$_4$-m | CSI$_{16}$-m | HSS-m |
> |:----|:----:|:----:|:---:|:-----:|:------:|:----:|
> | LDCast  | 0.6776 | 0.2476 | 0.1558 | 0.2096 | 0.3675 | 0.2375 |
> | DiffCast | 0.6986 | 0.1854 | 0.3111 | 0.3799 | 0.5303 | 0.4372 ||
> | STLDM | **0.7097** | **0.1840** | **0.3378** | **0.4137** | **0.5665** | **0.4671** |
>
> From the results in the table above, we conclude that even on a high-resolution dataset, the proposed SLTDM consistently achieves the best performance across all baselines. This supports our claim that STLDM can outperform existing methods regardless of image resolution.
>
> The result in included in the Appendix.
>
> **Requested Changes**
>
> > **R1**: Tone down ...
>
> As discussed in Section 4.2, we compared STLDM’s performance with the baselines.  While STLDM does not have the dominant performance across every dataset and metric, we note that its improvements (ranging from 0.31% to 8.06% across 14 metrics) are considerably greater than its degradations (ranging from 1.40% to 2.17% across 4 metrics). In addition, STLDM offers a more efficient way for both training and inference.
>
> > **R2**: Expand discussion of limitations ...
>
> To examine scalability to high-resolution datasets, we measured the inference time (in seconds) on the HKO-7 benchmark at three spatial scales (128, 256, and 512) using a single RTX3090 GPU.
>
> | Model | 128 | 256 | 512 |
> |:----|:------:|:------:|:------:|
> | LDCast | 4.76 | 12.17 | 46.69 |
> | PreDiff | 70.80 | 222.73 | 900.15 |
> | DiffCast | 20.50 | 25.06 | 99.90 |
> | STLDM | 0.55 | 0.66 | 2.41 |
>
> > **R3**: Training complexity ...
>
> We report the training time of STLDM and two baselines: LDCast and DiffCast, on the HKO-7 benchmark with a batch size of 4 over 10k training steps, using a single RTX3090 GPU in the table above. This serves as an indicator of their training complexity and computational costs.
>
> From the table above in **W2**, training STLDM from end to end requires the least training time, compared to LDCast and DiffCast. Therefore, STLDM demonstrates its efficiency not only in inference but also in training.

---

> ### Author Response · Authors · 2025-09-25
> **Response to Reviewer LXNm**
>
> > **R4**: Justification on the constraint loss. $\mathcal{L}_C$
>
> Without the constraint loss, the Translator, $\Psi_\theta$, is only implicitly constrained by the KL-divergence and diffusion loss mentioned in Terms C and E, respectively. As a result, the first estimation is not well regularised and often degenerates into meaningless noise, as shown in Figure 7. These noisy first estimations do not provide useful help to the Latent Denoising Network, $D_\theta$, resulting in less accurate final predictions.
>
> The role of $\mathcal{L}_C$ is to explicitly constrain the first estimation predicted by $\Psi\_\theta$  so that it becomes a meaningful conditional variable, effectively guiding $D\_\theta$ towards more accurate predictions. From Figure 7, the final prediction has a better spatial alignment with the first estimation when the constraint loss, $\mathcal{L}_C$, is applied. This improvement is further validated by the result in Table 2, which shows that incorporating $\mathcal{L}_C$ during the training significantly enhances forecasting skill scores.
>
> > **R5**: Discuss on interpretability and potential use
>
> With its fast inference speed and ability to produce realistic and sharp predictions, STLDM enables probabilistic nowcasting with large ensemble sizes in real-time forecasting operations. This allows weather forecasters to issue timely warnings and alerts to the public. Furthermore, it facilitates the assessment of the likelihood and alternative scenarios regarding the movement and severity of high-impact rainstorm events, thereby mitigating potential societal losses of life and property.

---

### Review · Reviewer_WV7f · 2025-09-10

**Summary Of Contributions:**

The paper splits the precipitation nowcasting task into two parts, first predicting 'global' structure in the image and then conditioning on that to predict the 'local' structure. This address problems in existing approaches which tend to either get reasonable global structure (while producing blurry local structure) or get plausible local structure (while predicting poorly due to missed global structure). The paper proposes a latent diffusion model, trained end-to-end, to jointly solve the parts of the task. The method achieves state of the art performance compared to a reasonable selection of other methods and is evaluated across a number of real datasets.

**Audience:**

Yes

**Audience Explanation:**

Some members of the TMLR audience work on this exact problem (nowcasting precipitation), and there are even more working on related topics (general forecasting and learning from time-series data) who might find the paper interesting.

**Claims And Evidence:**

Yes

**Claims Explanation:**

Claims of performance and good runtime are supported by ample experimental results on real data, compared to a decent selection of existing methods, and through ablation studies.

**Requested Changes:**

I have one critical request:
1. Are all the values reported in the tables the result of one run? It would improve credibility of the analysis to instead perform multiple runs and report the mean/median and SD/CI. Or is this not computationally feasible?

And here's a list of suggestions that I think would improve the presentation:
1. elaborate upon the difference between "more effective" and "superior to" in the last sentence of the abstract (or remove the redundancy if they mean the same thing)
2. writing "ie." isn't standard, i.e., it's usually rather written in the preceding manner, with two commas and two periods
3. when introducing notation, X, it helps to have a comma on both sides
4. in Section 2.2:
    1. singular "...a likelihood..." and then plural conjugation "...which do not suffer..."
    2. "AFNO" is undefined
    3. "stochastic" is an adjective, and its noun seems to be missing
5. page 6: I don't understand the sentence "Therefore, to enable..."; maybe it could be structured better to improve clarity?
6. page 7: "while without" could be changed to just "without"
7. the Limitation and future work section could use some polishing:
    1. maybe the plural "Limitations..." is more suitable
    2. having both "Although" and "however" in the first sentence is a bit redundant and confusing
    3. "multi-modal" is an adjective, and the noun its describing seems be missing (maybe "model"?)
    4. "even though" and "but" in the second-to-last sentence is redundant/confusing; maybe "that" missing between "events" and "happen"?
8. I found some of the word choices throughout the paper to a bit confusing/not quite right:
    1. page 2: mitigate, incorporation
    4. page 3: demonstrating, panacea, prominent, demonstrate

---

> ### Author Response · Authors · 2025-09-25
> **Response to Reviewer WV7f**
>
> We would like to thank the Reviewer WV7f's feedback and suggestions. We have modified our manuscript accordingly and highlighted those changes in blue. Here are our responses to each of Reviewer WV7f's concerns.
>
> **Requested Changes**
> > **Critical Request**: Are all the values reported in the tables the result of one run ...
>
> As mentioned in Section 4.2, the values reported in the tables are evaluated using 10 ensemble predictions (i.e., 10 runs). For those forecasting skill scores, we followed the approach used in the PreDiff work. For example, $\text{CSI} = \frac{Hits}{Hits + F.Alarm + Miss}$, where the counts of Hits (truth = 1, pred = 1), F.Alarm (truth = 0, pred = 1), and Miss (truth = 1, pred = 0) are accumulated over 10 ensemble predictions after binarising the predictions and truth.
>
> > **R1**: elaborate upon the difference ...
>
> We have modified it. The term "more effective" here is to indicate that STLDM requires a shorter sampling inference time, while "superior" means it achieves better quantitative and qualitative result.
>
> > **R2** and **R3**
>
> We have updated the writings of the symbol notations and "i.e.,".
>
> > **R4**
>
> We have updated the writing in Section 2.2 as highlighted in blue.
>
> > **R5**
>
> It has been changed to "To ensure that final ..." on page 6.
>
> > **R6**
>
> It have been updated as "... while maintaining ..." on page 7.
>
> > **R7**
>
> We have updated Section 5 - Conclusion and Future Work
>
> > **R8**
>
> We have replaced "mitigate" and "incorporation" on page 2 with "employ" and "cooperation" respectively, as highlighted in blue. Besides that, the words: "demonstrating", "panacea", "prominent", and "demonstrate" on page 3 are also replaced with "achieving", "reduces the heavy demand ...", "state-of-the-art" and "achieved"

---

### Review · Reviewer_4Mk9 · 2025-09-14

**Summary Of Contributions:**

This paper applies standard latent diffusion model on Nowcasting problem. The latent diffusion consists of a VAE and a latent denoising network. It claims the state of the art results on several nowcasting dataset.

**Audience:**

Yes

**Audience Explanation:**

While is latent diffusion model is standard, the application to nowcasting dataset could be interesting to some audience in the venue.

**Broader Impact Concerns:**

No ethical concerns

**Claims And Evidence:**

Yes

**Claims Explanation:**

The paper claims the proposed method is more effective and superior to the state of the art. According to Table 1, the propsed method (STLDM) has better accuracy compared to previous methods such as DiffCast. However, several previous works have explored generative models in related tasks such as CorrDiff, GenCast, GenFocal.

**Requested Changes:**

Diffusion models have been broadly applied in weather forecast, nowcasting, and downscaling. The author should discuss these previous work in the field:
- Corrdiff: Mardani, Morteza, et al. "Residual diffusion modeling for km-scale atmospheric downscaling." (2023).
- GenCast: Price, Ilan, et al. "Gencast: Diffusion-based ensemble forecasting for medium-range weather." arXiv preprint arXiv:2312.15796 (2023).
- GenFocal: Wen et al. "Regional climate risk assessment from climate models using probabilistic machine learning."

---

> ### Author Response · Authors · 2025-09-25
> **Response to Reviewer 4Mk9**
>
> We would like to thank the Reviewer 4Mk9's feedback and suggestions. We have modified our manuscript accordingly and highlighted those changes in blue. Here are our responses to each of Reviewer 4Mk9's concerns.
>
> **Requested Changes**
> > **R1**: Include previous work.
>
> Thanks for the suggestion of related works; we have added them in Section 2.2.

---

> > ### Comment · Reviewer_4Mk9 · 2025-10-27
> >
> > Thank the authors for adding the related works. I have some further questions:
> > - The Classifier Free Guidance is mentioned in Sec 3.1.2. However it is unclear whether (and how) CFG is applied to the model.
> > - For many latent diffusion models on video or time series, the forecasting step is done within the condition diffusion. But in this paper, there is a determinist translator "\Psi" doing the forecast, which is corrected by the diffusion model. It will be helpful to do an ablation where there are encoder-decoder and diffusion, but not the translator.
> > - Still the baselines in the paper seem not strong enough. It will be helpful to compare to at least of one the larger scale model such as the CorrDiff, GenCast, or Genfocal.

---

> > > ### Author Response · Authors · 2025-11-03
> > > **Response to Reviewer 4Mk9**
> > >
> > > We would like to thank Reviewer 4Mk9's follow-up feedback and concerns. Here are our responses to each of Reviewer 4Mk9's concerns.
> > >
> > > **Questions**
> > >
> > > > **Q1**: Whether and how CFG is applied to the model
> > >
> > > Yes, we apply Classifier-Free Guidance (CFG) in our model. Following prior work, we adopt a joint training strategy where the conditioning signal is randomly dropped with the probability of 15% (i.e., 85% conditional and 15% unconditional training). The unconditional case is implemented by replacing the conditioning input (i.e., the first estimation, $\bar{z}_{1:N}$) with a null embedding, enabling the model to learn both the conditional and unconditional score estimation jointly.
> > >
> > > During inference, at each denoising timestep, we compute both the conditional and unconditional noise predictions, which is two forward passes per step. We then combine these predictions and modify the denoising score according to Equation 6 with guidance strength, $w=1.0$. This ensures that CFG is applied throughout the sampling process.
> > >
> > > > **Q2**: An additional ablation study where there are encoder-decoder and diffusion modules, only with the removal of the Translator module
> > >
> > > Thank you for the suggestion. We conducted the ablation study to evaluate STLDM in the absence of the Translator module. In this setting, forecasting is performed directly by the diffusion model, where we concatenate the past frames before the noisy future frames so that it can denoise the future sequence based on the past. There is no deterministic forecasting module in this variant, and we reported the results on the SEVIR dataset as shown below.
> > >
> > > | Model | SSIM | LPIPS | CSI-m | CSI$_4$-m | CSI$_{16}$-m | HSS-m |
> > > |:----|:----:|:----:|:---:|:-----:|:------:|:----:|
> > > | STLDM w/o Translator |0.4812 | 0.4088 | 0.0083 | 0.0242 | 0.0915 | 0.0073 |
> > > | STLDM |**0.7183** | **0.1929** | **0.3804** | **0.4662** | **0.6178** | **0.5024** |
> > >
> > > From the result above, removing the Translator leads to a massive performance degradation across all metrics, especially in the forecasting skills: CSI and HSS. This indicates that the Translator, together with the constraint loss, $\mathcal{L}_{C}$, provides a good initialization for the future frames, which is then refined by the diffusion model. In other words, STLDM’s two-stage formulation: forecasting followed by visual enhancement, is crucial for accurate precipitation nowcasting.
> > >
> > > > **Q3**: More large-scale baselines, such as CorrDiff, GenCast, and Genfocal
> > >
> > > Thank you for the suggestion. CorrDiff, GenCast, and GenFocal are proposed for the medium-range forecasting (typically 3-10 days), whereas our work focuses on precipitation nowcasting over a short period (several hours). These models rely on the global atmospheric states, and are trained on medium-range climate datasets. In contrast, our task and benchmarks target the short-term convective precipitation.
> > >
> > > Due to differences in forecasting period, data scale, and data modality, directly applying these medium-range systems on this short-term nowcasting task would not yield a meaningful or fair comparison. Instead, we compare STLDM against those established nowcasting baselines (i.e., LDCast, PreDiff, and DiffCast), consistent with prior work in this domain.
> > >
> > > We appreciate the suggestion and believe that the extending STLDM towards large-scale forecasting frameworks is an interesting direction for our future work.

---

> > > > ### Comment · Reviewer_4Mk9 · 2025-11-03
> > > >
> > > > Thanks the author for the response.
> > > > Regarding CFG, in the convention, CFG is originally applied to text or class label (as in image generation), where the condition is a discrete token that can be easily masked out. On the other hand, here the conditioning input is the image output from the translator model. While there is some work that appled CFG at the image level, it has not been very standard. It could be helpful to add some details on how to do null embedding for images and add a table for ablation.

---

> > > > > ### Author Response · Authors · 2025-11-07
> > > > > **Response to Reviewer 4Mk9**
> > > > >
> > > > > Thank you for the comment. To apply Classifier-Free Guidance (CFG) in our STLDM, we concatenate the latent output from the Translator module, $\bar{z}\_{1:N}$, with the noisy latent input, $z_{1:N}^{t}$, along the channel dimension [1] at the first ResBlock of each Downsampling Block,  as illustrated in Figure 2. This corresponds to the conditional case. For the unconditional case, we replace  $\bar{z}_{1:N}$ with a zero tensor of the same size, which serves as the null embedding here.
> > > > >
> > > > > To validate CFG's effectiveness, we conducted an ablation study on the SEVIR dataset. As reported in the table below, applying the CFG technique consistently improves the model performance across all evaluation metrics.
> > > > >
> > > > > | Model | SSIM | LPIPS | CSI-m | CSI$_4$-m | CSI$_{16}$-m | HSS-m |
> > > > > |:----|:----:|:----:|:---:|:-----:|:------:|:----:|
> > > > > | w/o CFG |0.7041 | 0.2129 | 0.3719 | 0.4376 | 0.5614 | 0.4898 |
> > > > > | with CFG | **0.7183** | **0.1929** | **0.3804** | **0.4662** | **0.6178** | **0.5024** |
> > > > >
> > > > > These results demonstrate that incorporating CFG provides consistent performance improvements, indicating its effectiveness.
> > > > >
> > > > > [1] Prakhar Srivastava, Ruihan Yang, Gavin Kerrigan, Gideon Dresdner, Jeremy J McGibbon, Christopher S. Bretherton, and Stephan Mandt. Precipitation downscaling with spatiotemporal video diffusion. In NeurIPS, 2024.

---

> > > > > > ### Comment · Reviewer_4Mk9 · 2025-11-07
> > > > > >
> > > > > > Thanks the author for the response. My questions have been addressed.

---

### Decision · Action_Editor_ga9y · 2025-11-16

**Recommendation:** Accept with minor revision

**Additional Comments:**

The paper proposes STLDM, an end-to-end pipeline for nowcasting meteorological variables using a coarse-to-fine approach that combines a variational autoencoder with a latent diffusion model. Reviewers appreciated the evaluation, relevance, and timeliness of the work but also raised some concerns, including missing related work, lack of baseline comparisons/ablation studies, and questions regarding scalability and efficiency. The authors have addressed these issues effectively, and all three reviewers either recommend acceptance or are leaning in that direction. I share this view but request that the authors make the following minor revisions:

- Abstract: Revise to explicitly mention that the model is based on diffusion and clarify the role of the conditioning network.
- Background: Improve self-containedness. The diffusion loss appears to be missing an expectation, and the classifier-free guidance equation is difficult to follow without consulting the original paper.
- Figure 2: Rework to include \bar{z}_{1:N} in the top part of Figure 2 and clarify the relationship between the yellow box and green box.
- Experiments: Add the ablation study related to classifier-free guidance to (the appendix of) the paper.

Once these changes have been made, the paper will be recommended for acceptance.

**Audience:**

Yes

**Audience Explanation:**

Forecasting and nowcasting are fundamental tasks in meteorology, with applications in disaster management, aviation, and marine operations. Variational autoencoders have long been a foundation for latent variable modeling, and their combination with modern diffusion models is promising, timely, and of interest to the community. While the concept of coarse-to-fine prediction over multiple stages is not new, the specific setup proposed in this paper is a sensible variant of a proven concept. The efficiency of STLDM may encourage broader adoption.

**Claims And Evidence:**

Yes

**Claims Explanation:**

The paper provides an adequate discussion of related work, including an overview of recent diffusion models and their applications in forecasting and nowcasting. The experimental evaluation includes baseline comparisons against three recent diffusion models (LDCast, PreDiff, and DiffCast) and four deterministic models (ConvLSTM, PredRNN, SimVP, and Earthformer) on three datasets (SEVIR, HKO-7, and MeteoNet). Using widely adopted forecasting metrics such as the Critical Success Index (CSI) and the Heidke Skill Score (HSS), along with perceptual metrics including the Structural Similarity Index Measure (SSIM) and the Learned Perceptual Image Patch Similarity (LPIPS), STLDM is shown to outperform the baselines in most experimental settings.